# BoreaRL: A Multi-Objective Reinforcement Learning Environment for Climate-Adaptive Boreal Forest Management

**Kevin Bradley Dsouza**[1]    **Enoch Ofosu**[1]    **Daniel Chukwuemeka Amaogu**[2]

**Jérôme Pigeon**[2]    **Richard Boudreault**[1,3]    **Pooneh Maghoul**[2]    **Juan Moreno-Cruz**[1]

**Yuri Leonenko**[1]

[1]University of Waterloo    [2]Polytechnique Montréal    [3]Université de Sherbrooke

[1]{k5dsouza, e2ofosu, juan.moreno-cruz, leonenko}@uwaterloo.ca
[2]{daniel-chukwuemeka.amaogu, jerome.pigeon}@etud.polymtl.ca
[2]pooneh.maghoul@polymtl.ca
[3]Richard.Boudreault@usherbrooke.ca

## Abstract

Boreal forests store 30-40% of terrestrial carbon, much in climate-vulnerable permafrost soils, making their management critical for climate mitigation. However, optimizing forest management for both carbon sequestration and permafrost preservation presents complex trade-offs that current tools cannot adequately address. We introduce BoreaRL, the first multi-objective reinforcement learning environment for climate-adaptive boreal forest management, featuring a physically-grounded simulator of coupled energy, carbon, and water fluxes. BoreaRL supports two training paradigms: site-specific mode for controlled studies and generalist mode for learning robust policies under environmental stochasticity. Through evaluation of multi-objective RL algorithms, we reveal a fundamental asymmetry in learning difficulty: carbon objectives are significantly easier to optimize than thaw (permafrost preservation) objectives, with thaw-focused policies showing minimal learning progress across both paradigms. In generalist settings, standard gradient-descent based preference-conditioned approaches fail, while a naive site selection approach achieves superior performance by strategically selecting training episodes. Analysis of learned strategies reveals distinct management philosophies, where carbon-focused policies favor aggressive high-density coniferous stands, while effective multi-objective policies balance species composition and density to protect permafrost while maintaining carbon gains. Our results demonstrate that robust climate-adaptive forest management remains challenging for current MORL methods, establishing BoreaRL as a valuable benchmark for developing more effective approaches. We open-source BoreaRL to accelerate research in multi-objective RL for climate applications.

## 1 Introduction

Boreal forests are one of the largest terrestrial biomes, circling the Northern Hemisphere and storing an estimated 30-40% of the world's land-based carbon, much of it in permafrost soils (Bradshaw & Warkentin, 2015). These ecosystems overlay vast regions of permafrost, carbon-rich frozen ground that is highly susceptible to climate warming (Schuur et al., 2015). The release of this permafrost carbon pool poses a significant risk of a positive feedback to global warming. As such, the stewardship of boreal regions is not merely a regional concern but a global climate imperative.

Afforestation has emerged as an important nature-based climate solution (Drever et al., 2021), with the potential to sequester substantial atmospheric carbon while providing co-benefits for biodiversity and ecosystem services. In boreal regions specifically, afforestation strategies can play a dual role in climate mitigation: directly removing carbon dioxide from the atmosphere through enhanced vegetation growth, and indirectly protecting vast soil carbon stores by preventing permafrost thaw (Heijmans et al., 2022; Stuenzi et al., 2021). The unique characteristics of boreal ecosystems: their extensive permafrost coverage, extreme seasonal variability, and dominance by coniferous and deciduous species with contrasting biogeophysical properties, make them particularly promising targets for strategic afforestation efforts that can optimize for multiple objectives.

In the context of afforestation, a complex trade-off exists between maximizing carbon sequestration and maintaining permafrost stability through the surface energy balance, which is strongly modulated by forest structure through multiple interconnected pathways (Dsouza et al., 2025). Dense coniferous forests excel at carbon uptake due to their year-round photosynthetic activity, but their dark evergreen canopies have low albedo, absorbing more solar radiation during the growing season while intercepting winter snowfall. This creates competing effects: reduced ground snowpack allows cold winter air to penetrate the soil (benefiting permafrost), but large summer energy gains can overwhelm these winter benefits (Heijmans et al., 2022). In contrast, deciduous forests allow deep snowpack formation that insulates the soil (potentially accelerating thaw), yet their higher albedo when leafless reduces energy absorption during spring periods (Heijmans et al., 2022). These counteracting biogeophysical effects create a complex optimization landscape where the ideal management strategy depends on interactions between climate, species composition, stand density, and management timing, and the relative weighting of carbon versus permafrost objectives (Bonan, 2008).

Developing prescriptive strategies for this multi-objective problem requires tools that can discover optimal policies and not just predict outcomes. Reinforcement learning (RL) is uniquely suited to this challenge because it learns sequential policies through trial-and-error interaction with complex environments (Sutton et al., 1998), handling the long-term consequences and delayed multi-decadal rewards inherent in forest management. Agents must navigate a non-convex landscape of conflicting objectives (carbon vs. thaw), generalize across diverse stochastic site conditions, and solve a challenging long-horizon credit assignment problem where early management decisions determine permafrost outcomes decades later. Moreover, the inherently multi-objective nature of forest management, where carbon sequestration, permafrost preservation, and potentially other goals like biodiversity or economic returns must be simultaneously optimized, makes this an ideal application domain for Multi-Objective Reinforcement Learning (MORL) (Roijers et al., 2013; Liu et al., 2014).

However, the application of RL to climate-adaptive forest management has been hindered by the lack of suitable training environments that combine modern RL algorithms with realistic physics simulation and explicit multi-objective optimization capabilities. Existing forest models are designed for prediction rather than policy optimization, while previous RL approaches have relied on oversimplified growth models that cannot capture the complex biogeophysical trade-offs critical to climate adaptation. Our contributions address this gap and are fourfold:

1. We introduce BoreaRL, a configurable multi-objective RL environment and framework for boreal forest management. BoreaRL provides a physically-grounded simulator of coupled energy, carbon, and water fluxes with modular components for physics, rewards, agents, and evaluations, enabling the first systematic study of afforestation trade-offs in a MORL setting.

2. We design and validate two distinct training paradigms: site-specific mode for controlled studies and generalist mode for robust policy learning under environmental stochasticity. We reveal a fundamental asymmetry in learning difficulty, where carbon objectives are easier to optimize than thaw objectives, with standard preference-conditioned approaches failing in generalist settings.

3. We demonstrate that even naive baselines like adaptive episode selection outperforms standard MORL approaches in generalist settings, achieving superior empirical trade-off coverage. Our analysis of emergent strategies learned by the RL policies reveals distinct approaches to density and species composition management that reflect the trade-offs between carbon and permafrost.

4. Our framework provides a principled testbed for MORL in physically-grounded domains that isolates asymmetric difficulties, generates testable scientific hypotheses, and offers a high-impact platform for addressing the existential threat of climate change.

## 2 RELATED WORK

**Multi-Objective Reinforcement Learning (MORL):** MORL extends the RL framework to handle problems that involve multiple conflicting objectives by learning policies that can navigate these trade-offs (Roijers et al., 2013; Liu et al., 2014; Roijers et al., 2018). A common approach is linear scalarization, where the vectorized reward is reduced to a scalar using fixed weights, which requires training a new agent for every desired trade-off (Vamplew et al., 2011). More advanced methods, such as preference-conditioned RL, aim to learn a single, generalist policy $\pi(a|s, w)$ that can adapt its behavior based on a given preference vector $w$ (Mu et al., 2025; Abels et al., 2019). Recent work has also adapted modern policy gradient methods like Proximal Policy Optimization (PPO) (Schulman et al., 2017) for multi-objective settings (Hayes et al., 2021). We demonstrate the utility of our BoreaRL environment using multiple methods from these families as baselines.

**Reinforcement Learning for Environmental Management:** RL has been applied to forest management and environmental conservation before (Malo et al., 2021; Bone & Dragićević, 2010; Overweg et al., 2021; Lapeyrolerie et al., 2022), however, these approaches relied on simplified growth models, static datasets, or adjacent domains. BoreaRL implements a more realistic environment for training RL algorithms using coupled energy-water-carbon flux simulator with implicit energy balance equations, detailed snow dynamics, and climate-driven uncertainty, enabling more accurate representation of complex ecological trade-offs in boreal systems.

**Process-Based Forest Models:** Simulating forest dynamics has evolved into sophisticated process-based ecosystem models like CLASSIC (Melton et al., 2020) and the Community Land Model (CLM5) (Fisher et al., 2019), which represent vegetation responses to biogeochemical forcing through parametric controls that govern plant physiology, carbon allocation, and nutrient cycling. The Canadian Forest Service's Carbon Budget Model (CBM-CFS3) extends this paradigm by explicitly incorporating forest management decisions and natural disturbances (Kurz et al., 2009). Our work combines process-based modeling with learning management decisions under one roof, creating the first modular, scientifically credible RL environment that seamlessly integrates detailed ecosystem simulation with modern RL frameworks.

## 3 THE BOREARL ENVIRONMENT AND FRAMEWORK

BoreaRL is designed as a modular, configurable framework with plug-in components for different aspects of the multi-objective forest management problem (Fig. 1). The core architecture consists of a physically-grounded simulator (*BoreaRL-Sim*) and a flexible MORL environment wrapper (*BoreaRL-Env*) that supports multiple training paradigms and evaluation protocols, conforming to the *mo-gymnasium* API standard (Felten et al., 2023).

Our framework provides compartmentalized modules for: (1) *Physics simulation* with selectable backend and temporal resolution; (2) *Reward specification* with customizable objective functions and normalization schemes; (3) *Agent interfaces* supporting both single-policy and multi-policy MORL algorithms; (4) *Environmental stochasticity* with controllable weather generation and parameter sampling; and (5) *Evaluation protocols* with standardized metrics for multi-objective assessment. The framework also allows users to override default parameters for physics simulation, reward shaping, observation space structure, and training protocols. We envision that this design will enable researchers to solve a wide array of forest management problems and build custom agents.

### 3.1 BOREAL FOREST SIMULATOR (BOREARL-SIM)

The first part of our framework is a process-based simulator that models coupled energy, water, and carbon fluxes on a $n$-minute time step. The simulator incorporates several key components: a multi-node energy balance model spanning canopy, trunk, snow, and soil layers; a dynamic carbon cycle; a comprehensive water balance that includes snow dynamics; and stochastic modules for fire and insect disturbances that are conditioned on climate and stand state. The model uses standard, validated physical formulations found in major Land Surface Models like CLM5 (Fisher et al., 2019; Lawrence et al., 2019) and CLASSIC (Melton et al., 2020). The simulator operates in two distinct modes: generalist mode, where each $H$-year episode is driven by unique, stochastically generated weather sequences with site parameters sampled from realistic ranges to ensure policy robustness

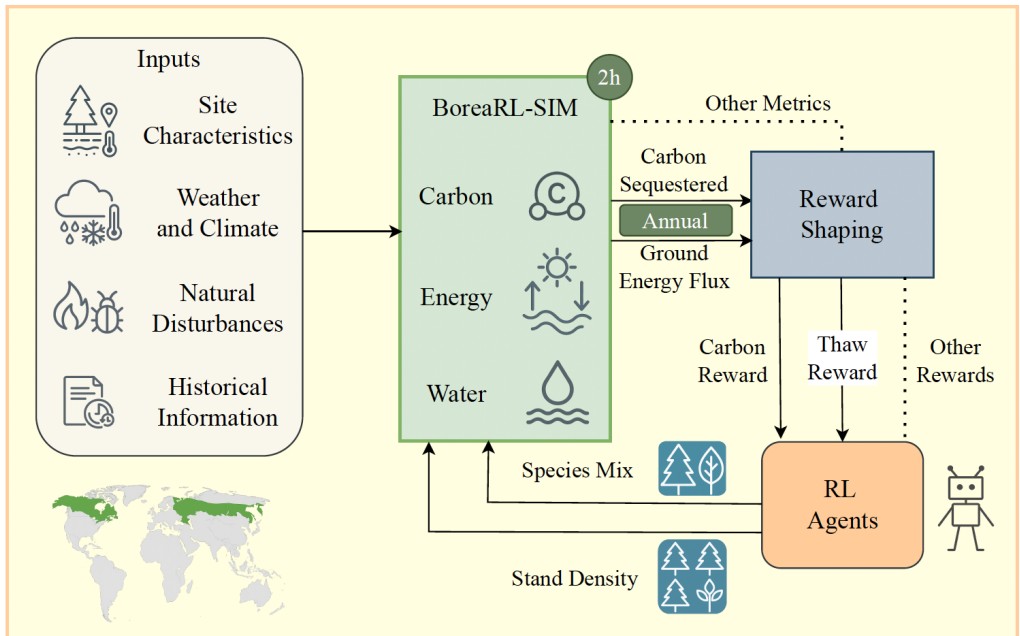

Figure 1: **BoreaRL environment**. A physics-aware boreal forest simulator (BoreaRL-SIM) consumes site characteristics, weather & climate, natural disturbances, and historical information, returning annual carbon and ground energy flux metrics. Reward shaping converts simulator outputs into learning signals. RL agents act on these rewards to learn annual policies of stand density and species mix that maximize long-term carbon while limiting permafrost thaw.

across diverse conditions; and site-specific mode, which uses deterministic weather patterns and fixed site parameters for reproducible, location-targeted optimization. For detailed descriptions of the simulator's physics, management, and disturbance implementations, see Appendix B.

## 3.2 MULTI-OBJECTIVE RL ENVIRONMENT (BOREARL-ENV)

We formalize the management task as a Partially Observable Markov Decision Process for MORL, defined by the tuple $(\mathcal{S}, \mathcal{A}, \mathcal{O}, P, \boldsymbol{R}, \gamma)$.

**Observation Space ($\mathcal{O}$):** In the *generalist mode*, the agent receives an observation vector designed for training robust policies across diverse forest sites. In *site-specific mode* observation dimensionality is reduced, with fixed weather patterns, and deterministic initial conditions, suitable for location-specific policy optimization. See observation space details in Table 1.

**Action Space ($\mathcal{A}$):** The discrete action space encodes two management dimensions into a single value: (1) *Stand Density Change* $\{-100, -50, 0, +50, +100\}$ stems ha$^{-1}$, representing thinning or planting, and (2) *Species Mix Change* representing the target conifer fraction $\{0.0, 0.25, 0.5, 0.75, 1.0\}$. The transition function takes an action at year $t$, updates the stand properties accordingly, and the simulator runs for one year to produce the next state at $t+1$.

**Vector Reward Function ($\boldsymbol{R}$):** To handle the multi-objective problem, the environment returns a reward vector at each step $t$, $\boldsymbol{R}_t = [r_{c,t}, r_{t,t}]$, where components are normalized to $[-1, 1]$.

The **Carbon Reward** ($r_{c,t}$) incentivizes carbon sequestration while penalizing ecological violations:

$$r_{c,t} = \text{clip}\left(\text{clip}\left(\frac{\Delta C_t}{2.0}, -1, 1\right) + s_b + h_b - p_{limit} - p_{density} - p_{ineff}, -1, 1\right) \quad (1)$$

where $\Delta C_t$ is the net ecosystem carbon change (kg C m$^{-2}$ yr$^{-1}$). $s_b$ and $h_b$ are bonuses for total stock and HWP storage. Penalties include $p_{limit}$ for exceeding realistic biomass ($> 15$ kg C m$^{-2}$)

| Component | Dimensions | Description |
|---|---|---|
| Current Ecological State | 4 | Year, stem density, conifer fraction, carbon stock |
| Site Climate Parameters | 6 | Latitude, mean annual temperature, seasonal amplitude, phenological dates, growing season length |
| Historical Information | 17 | History of disturbances, management actions, and carbon flux trends spanning recent years |
| Age Distribution | 10 | Fraction of stems in each age class (seedling-old) for both coniferous and deciduous species |
| Carbon Stock Details | 2 | Normalized biomass and soil carbon pools |
| Penalty Information | 3 | Indicators for ecological limit violations and management inefficiencies |
| Preference Input | 1 | Weight $w_C \in [0, 1]$ for the carbon objective, enabling preference-conditioned policies |
| Site Parameter Context | 62 | Episode-level site parameters sampled from physics model ranges (generalist mode only) |

Table 1: Observation space components in BoreaRL's generalist mode (105 dimensions) and site-specific mode (43 dimensions). The generalist mode includes episode-level site parameters for robust policy learning across diverse forest sites, while site-specific mode uses fixed parameters for location-targeted optimization.

or soil carbon ($> 20$ kg C m$^{-2}$) pools, $p_{density}$ for exceeding maximum stand density ($> 2000$ stems/ha), and $p_{ineff}$ for ineffective actions (e.g., thinning empty stands).

The **Thaw Reward** ($r_{t,t}$) is designed to protect permafrost by minimizing deep soil warming. It is calculated as an asymmetric function of the conductive heat flux to the deep soil layer:

$$r_{t,t} = \text{clip}\left( \frac{f_{cool} - \alpha \cdot f_{warm}}{40.0}, -1, 1 \right) \tag{2}$$

where $f_{cool}$ and $f_{warm}$ are the cumulative annual cooling and warming heat fluxes (MJ m$^{-2}$) across the permafrost boundary. The factor $\alpha = 2.5$ penalizes warming, reflecting the precautionary principle that permafrost degradation is often irreversible. We deliberately chose this physically-grounded reward based on soil heat flux rather than simpler proxies (like air temperature) to capture the complex, often delayed thermal inertia of permafrost soils. Both carbon and thaw rewards are constructed using existing common knowledge from literature about how these fluxes operate. Preference-conditioned training is supported through a preference weight input $w_C \in [0, 1]$ in the observation space. For more details on construction of the RL environment, see Appendix C.

### 3.3 TRAINING PARADIGMS AND MULTI-OBJECTIVE FORMULATION

**Site-specific vs. generalist settings:** Let $\phi \in \Phi$ denote a vector of episode-level site parameters that parameterize the transition kernel $P_\phi(s_{t+1} \mid s_t, a_t)$ and the vector reward $\mathbf{R}_\phi(s_t, a_t) = [R_{\text{carbon}}, R_{\text{thaw}}]^\top$. In the *site-specific* setting we fix $\phi = \phi_\star$, so the agent optimizes a single MDP/POMDP. In the *generalist* setting we sample $\phi \sim \mathcal{D}_{\text{site}}$ at the start of each episode from a known distribution over sites and climates; optionally, a context vector $\psi(\phi)$ is appended to the observation (Table 1). The resulting objective is a mixture-of-MDPs:

$$J(\pi) = \mathbb{E}_{\phi \sim \mathcal{D}_{\text{site}}} \mathbb{E}_{\tau \sim P_\pi^\phi} \left[ \sum_{t \geq 0} \gamma^t \mathbf{R}_\phi(s_t, a_t) \right],$$

where $P_\pi^\phi$ is the trajectory measure induced by $\pi$ and $P_\phi$.

We write the user preference as $\lambda = (w_C, w_P) \in \Delta^1$ with $w_C \in [0, 1]$ and $w_P = 1 - w_C$. Linear scalarization produces a scalar reward

$$r_t^\lambda = \lambda^\top \mathbf{R}_\phi(s_t, a_t) = w_C R_{\text{carbon},t} + (1 - w_C) R_{\text{thaw},t}.$$

Two regimes are useful here: (i) *fixed weight* $\lambda \equiv \bar{\lambda}$ (constant across training and evaluation), and (ii) *sampled weight* where $\lambda$ is provided as input to the policy.

### 3.4 Multi-Objective RL Baseline Algorithms

As a starting point, we implement and evaluate some simple multi-objective RL approaches that handle carbon-thaw trade-off differently.

**Fixed Lambda EUPG (Expected Utility Policy Gradient):** Fixed Lambda EUPG trains a single policy $\pi_\theta(a \mid o)$ on a scalarized reward using a fixed preference weight $\lambda$ throughout training. The scalarized reward is $r_t^\lambda = \lambda \cdot R_{carbon,t} + (1 - \lambda) \cdot R_{thaw,t}$. Following EUPG (Roijers et al., 2018), the policy input is augmented with the per-objective accrued return vector, enabling Expected Utility of the Returns (ESR)-consistent credit assignment. The objective is:

$$J_{\text{EUPG}}(\theta; \lambda) \;=\; \mathbb{E}_{\phi \sim \mathcal{D}_{\text{site}}} \; \mathbb{E}_{\tau \sim P_{\pi_\theta}^\phi} \left[ \sum_{t \geq 0} \gamma^t \, r_t^\lambda \right].$$

**Variable Lambda EUPG (Adaptive Preference Learning):** Variable Lambda EUPG trains a single policy $\pi_\theta(a \mid o)$ that learns to adapt to weights by sampling $\lambda \sim \mathcal{D}_\Lambda$ for each episode:

$$J_{\text{VarEUPG}}(\theta) \;=\; \mathbb{E}_{\lambda \sim \mathcal{D}_\Lambda} \; \mathbb{E}_{\phi \sim \mathcal{D}_{\text{site}}} \; \mathbb{E}_{\tau \sim P_{\pi_\theta}^\phi} \left[ \sum_{t \geq 0} \gamma^t \, r_t^\lambda \right].$$

The policy receives the preference weight as part of its observation space (Table 1) and learns to adjust its behavior accordingly, making it preference-conditioned. This approach enables a single policy to handle multiple trade-offs without retraining.

**PPO Gated (Proximal Policy Optimization with Action Masking):** PPO Gated implements a standard PPO algorithm with a policy $\pi_\theta(a \mid o)$ and a gated architecture that separates planting actions (positive density changes) from non-planting actions (thinning or no change), with separate neural network heads for each action type. The gating mechanism ensures that only valid actions are considered based on the current forest state, such as preventing planting when at maximum density or thinning when no old trees are available. The objective follows standard PPO.

## 4 Experiments

### 4.1 Experimental Design

For site-specific mode, we train 5 agents with different random site seeds. For generalist mode, a single agent is trained, and each episode utilizes a unique random seed to sample diverse weather conditions and site parameters, ensuring robustness. Reported results for generalist mode are averages over these 100 evaluation episodes. We vary the number of training steps depending on whether the experiment is site-specific or generalist, and whether its fixed-weight or sampled-weight. We compare against heuristic baselines (zero density change, +100 density increase with 0.5 conifer fraction, target density of 1000 stems/ha, and conifer restoration of 100% conifer) and evaluate performance using learning curves, reward metrics, and learned strategies of the RL agents.

### 4.2 Asymmetric Learning Difficulty in Carbon vs. Thaw Objectives

We find that there is an asymmetry in learning difficulty between the two objectives across both site-specific and generalist settings. Fixed-weight agents demonstrate that carbon objectives are easier to learn than thaw objectives, with thaw-preferred policies showing minimal or no learning progress in many cases. This asymmetric learning landscape emerges from the underlying physics: carbon rewards provide clear, immediate feedback through biomass accumulation, while thaw rewards depend on complex seasonal energy balance dynamics with delayed and noisy signals. This is exacerbated by the necessary risk-averse formulation of the thaw objective, which reflects that permafrost degradation is often irreversible and more damaging than equivalent cooling is beneficial.

Fig. 2 shows the scalarized reward learning curves for both generalist and site-specific settings, demonstrating this asymmetric learning pattern. In the generalist setting (Fig. 2a), carbon-focused

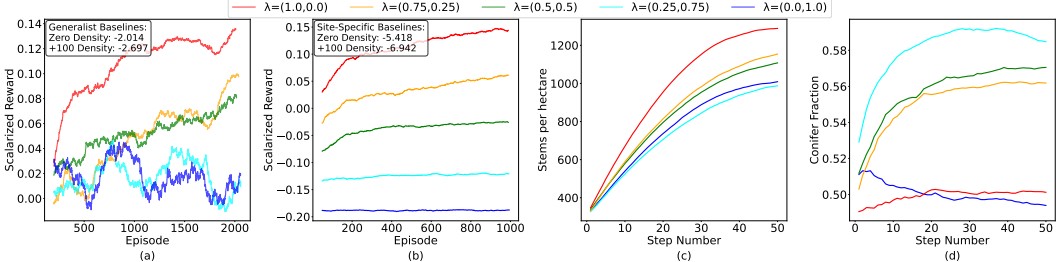

Figure 2: Asymmetric learning difficulty between carbon and thaw objectives. (a,b) Carbon-focused policies ($\lambda = (1.0, 0.0)$) achieve rapid learning while thaw-focused policies ($\lambda = (0.0, 1.0)$) show minimal improvement in both generalist and site-specific settings. (c) Carbon strategies favor aggressive density increases (1280 stems/ha) while thaw strategies remain conservative (1000-1020 stems/ha). (d) Species composition shows complex patterns: carbon policies maintain status quo, mixed policies achieve highest conifer fractions, and thaw policies promote deciduous dominance.

policies ($\lambda = (1.0, 0.0)$) achieve rapid learning, while thaw-focused policies ($\lambda = (0.0, 1.0)$) show minimal improvement, remaining near baseline performance. The site-specific setting (Fig. 2b) exhibits similar patterns. Nonetheless, distinct forest management strategies emerge from these experiments. Stem density evolution over training episodes (Fig. 2c) shows carbon-focused policies aggressively increase density to 1280 stems/ha, while thaw-focused policies maintain conservative densities around 1000-1020 stems/ha. Species composition strategies on the other hand do not follow a simple carbon-thaw dichotomy (Fig. 2d). While purely carbon-focused policies show minimal conifer fraction change, purely thaw-focused policies promote deciduous dominance. Though the emergence of these qualitatively distinct strategies suggests that agents develop preference-specific management approaches, no strong conclusion can be drawn about the thaw policies as they may be a conservative result of lack of learning. Nevertheless, we will later show that well balanced policies are a hallmark of agents that do learn as well.

### 4.3    CURRICULUM SITE SELECTION AND PERFORMANCE IN GENERALIST MODE

Given that thaw rewards are harder to learn (Section 4.2) and are majorly influenced by chosen sites (see Appendix D.1; Fig. 5), we wanted to check the impact simple site selection. To do so, we implement adaptive episode selection. Our Curriculum baseline allows the agent to ignore these destabilizing sites, and consolidate its policy on a "safer" subset (see Table 12).

**Curriculum PPO (Adaptive Episode Selection):**   Curriculum PPO implements a two-level decision process that combines episode selection with action selection. Like the other baselines, Curriculum PPO is also preference-conditioned (the preference weight $\lambda$ is included in the observation space), but differs in its training mechanism through adaptive episode selection. The agent first decides whether to train on a given episode using a curriculum selection network $f_\phi(o_{site}) \to [0, 1]$ that evaluates the episode's potential learning value based on site characteristics. Episodes are selected based on an adaptive threshold, and then they proceed with standard PPO action selection. The combined objective is:

$$J_{\text{Curriculum}}(\theta, \phi) \;=\; \mathbb{E}_{\lambda \sim \mathcal{D}_\Lambda} \, \mathbb{E}_{\phi \sim \mathcal{D}_{\text{site}}} \, \mathbb{E}_{select \sim f_\phi} \left[ \mathbb{E}_{\tau \sim P_{\pi_\theta}^\phi} \left[ \sum_{t \geq 0} \gamma^t \, r_t^\lambda \right] \mid select = 1 \right],$$

where the expectation is taken only over selected episodes. The curriculum selection network $f_\phi$ is an untrained random projection that provides a consistent ordering of the site space, while the selection threshold is adaptively adjusted based on performance to expand or contract the training distribution (see Appendix C.5 for details on the mechanism).

We consider preference-conditioned generalist mode to be our benchmark setting, and evaluate three preference-conditioned algorithms (Variable $\lambda$ EUPG, PPO Gated, and Curriculum PPO) in generalist mode to assess their ability to learn policies under environmental stochasticity. For detailed

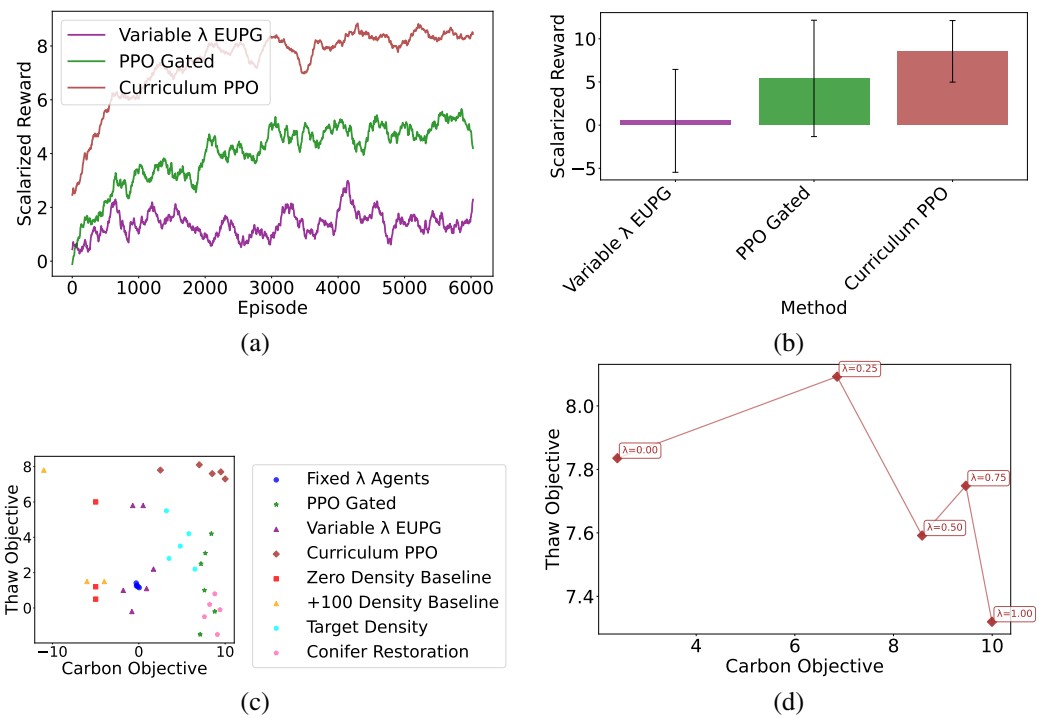

Figure 3: Algorithm performance comparison in generalist mode. (a) Learning curves reveal Curriculum PPO's rapid convergence and stable performance versus others. (b) Scalarized evaluation reward demonstrates Curriculum PPO's dominance, whereas Variable $\lambda$ EUPG has near-zero performance. Error bars represent the standard deviation over 100 evaluation episodes (per preference weight), each with a unique random seed. (c) Trade-off analysis shows the relationship between evaluation carbon and thaw objectives for different methods versus baselines. d) Curriculum PPO empirical trade-off coverage achieves superior spread with lesser $\lambda$-monotonicity violations compared to other methods. Shows mean over 100 evaluation episodes with unique random seeds. See Appendix D.2 for fronts of other methods. All rewards and objectives are summed over 50 steps.

runtime analysis, see Appendix C.9. Curriculum PPO baseline outperforms other methods across training (Fig. 3a) and evaluation (Fig. 3b) metrics (scalarized rewards). Moreover, Curriculum PPO produces the most comprehensive trade-off coverage (Fig. 3c,d), while others show poor performance across the preference space with higher $\lambda$-monotonicity violations (defined as the failure of an objective's return to increase with its preference weight; see Appendix D.2).

Table 2 compares our RL agents against heuristic baselines. While these fixed strategies achieve moderate scalarized rewards, they fail to capture the multi-objective front. We use hypervolume and sparsity as key metrics to quantify the quality and uniformity of the learned trade-off fronts, where Curriculum PPO demonstrates superior coverage (Hypervolume: 84.3). We also experiment with altenate thaw reward formulations. Table 3 reveals that while the Asymmetric Thaw formulation is the most challenging to optimize due to its risk-averse nature, Contrast Thaw and Raw Degree Days yield only marginally better scalarized rewards and lack the necessary penalty for irreversible permafrost degradation. Moreover, the Asymmetric formulation forces strong avoidance of warming, whereas symmetric formulations allow small warming trade-offs (see Table 10). All three formulations create conflicts with the carbon objective, as the physical mechanisms that benefit carbon often harm permafrost (see Appendix D.3, Table 11 for some mechanistic evidence). omparison of Thaw Reward Formulations and Agent Behavior. For a detailed breakdown of carbon and thaw objectives with error estimates across all RL baselines, see Table 13 in Appendix D.4.

Curriculum PPO's success highlights that the *existence* of a curriculum (selective exposure) already helps; while an *optimal ordering* of that curriculum may further improve performance, our results show that even a simple adaptive threshold provides significant benefits. This serves as a naive

Table 2: Main performance comparison of multi-objective RL algorithms and baselines. Rewards are averaged and other metrics are computed over 100 evaluation episodes per preference weight. Scalarized reward is the primary training objective. Hypervolume (reference point: $[-2, -2]$) and Sparsity measure the quality and uniformity of the trade-off front.

| Method | Scalarized Reward | Hypervolume ($\uparrow$) | Sparsity ($\downarrow$) |
|---|---|---|---|
| *RL Algorithms (Generalist Mode)* | | | |
| Curriculum PPO | $8.5 \pm 3.0$ | 84.3 | 0.12 |
| PPO Gated | $4.7 \pm 6.0$ | 23.6 | 0.09 |
| Variable $\lambda$ EUPG | $1.7 \pm 5.0$ | 14.2 | 0.07 |
| *Heuristic Baselines* | | | |
| Target Density (1000 stems/ha) | $4.3 \pm 3.4$ | 20.6 | N/A |
| Conifer Restoration (100% Conifer) | $4.1 \pm 2.9$ | 21.4 | N/A |
| Zero Density Change | $-2.5 \pm 2.4$ | 11.3 | N/A |
| +100 Density Change | $-3.2 \pm 6.1$ | 18.5 | N/A |

Table 3: Impact of thaw reward formulation on PPO Gated performance. We compare the default Asymmetric Thaw (risk-averse) against Contrast Thaw (symmetric) and Raw Degree Days (linear). All three formulations are hard to optimize with a naive baseline, pointing to the difficulty of mastering the thaw regime. Asymmetric Thaw is the hardest, but also ecologically safer.

| Formulation | Scalarized | Carbon | Thaw | Hypervol. ($\uparrow$) | Sparsity ($\downarrow$) |
|---|---|---|---|---|---|
| Asymmetric (Default) | $4.7 \pm 6.0$ | $7.8 \pm 2.5$ | $1.5 \pm 1.2$ | 23.6 | 0.09 |
| Contrast | $4.9 \pm 3.2$ | $7.6 \pm 2.6$ | $2.1 \pm 2.4$ | 25.4 | 0.08 |
| Raw Degree Days | $5.2 \pm 3.7$ | $7.9 \pm 2.4$ | $2.4 \pm 2.3$ | 26.1 | 0.08 |

baseline to demonstrate that effective episode and site selection is one way to succeed in preference-conditioned learning in our BoreaRL environment. It also points to the fact that, certain sites and settings are inherently bad for planting when multiple objectives are important, no matter what the management decision are, and that smart site selection is crucial. That said, we believe that this benchmark is far from saturated and various other properties of the physics, objectives, simulation, and real-world can be exploited to train better RL management agents.

## 4.4 COMPARATIVE ANALYSIS OF MANAGEMENT STRATEGIES

To understand the learned behavior of different algorithms, we analyze their strategies (Fig. 4). Three distinct philosophies emerge from our analysis. PPO Gated develops an aggressive carbon-maximizing approach, achieving the highest conifer fractions when averaged across all evaluation episodes and weights (Fig. 4a) through rapid early planting followed by natural thinning (Fig. 4b). This strategy concentrates forest outcomes in high-density, high-conifer regions (Fig. 4c). In contrast, Curriculum PPO learns a balanced approach, showing steady species optimization while maintaining moderate density strategies. Variable $\lambda$ EUPG adopts the most conservative strategy, maintaining baseline species composition and growth, suggesting limited learning (Fig. 4a,b,c).

The environmental implications of these management strategies are revealed through the correlation between growing season length and thaw rewards (Fig. 4d). The difference in actions reflects a fundamental difference in the quality of the local optima found by each algorithm (see Table 9). PPO Gated falls into a local optimum of aggressive carbon farming (high density/conifer), which boosts carbon but fails to protect permafrost. Curriculum PPO maintains moderate densities and mixed species, allowing for longer growing seasons (see Fig. 4d), which enhances canopy shading and transpiration, thereby reducing soil heating and preventing thaw. We observe a strong correlation between growing season length and thaw protection ($r = +0.65$, see Table 8). Mechanistically, longer growing seasons maintain high Leaf Area Index (LAI) for more days, increasing transpiration

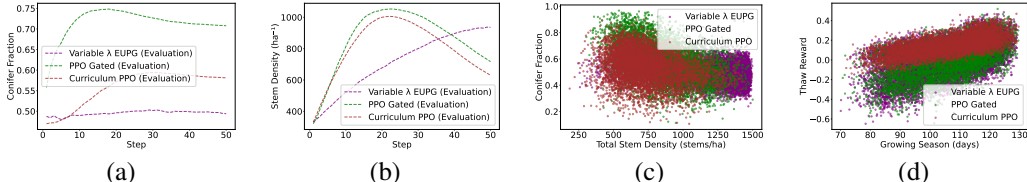

Figure 4: Comparative analysis of management strategies. Averaged across all evaluation episodes and weights. (a) Species composition evolution shows PPO Gated achieving highest conifer fractions, Curriculum PPO demonstrating steady improvement, and Variable $\lambda$ EUPG being conservative. (b) Density evolution reveals rapid early growth for PPO Gated and Curriculum PPO versus linear growth for Variable $\lambda$ EUPG. (c) Forest composition shows PPO Gated favoring high-density coniferous stands, Curriculum PPO achieving balanced mid-range strategies, and Variable $\lambda$ EUPG not converging to any approach. (d) Growing season vs. thaw reward correlation shows how longer growing seasons can enhance permafrost protection. Mechanisms like increased shading and evapotranspiration may play a role, see Table 8 for more.

cooling ($r = +0.82$) and canopy shading ($r = -0.75$). Therefore, strategically managed forests can serve as buffers against climate-induced permafrost degradation.

Moreover, these distinct physical strategies are driven by specific algorithmic failure modes. PPO Gated suffers from consistent gradient from the Carbon objective overpowering the noisy, sparse Thaw signal, leading the agent to ignore the latter. Variable $\lambda$ EUPG sees conflicting gradients from changing preference weights, leading to risk-averse inaction (low density, minimal changes). Curriculum PPO overcomes these issues through gradient filtering: by adaptively selecting sites where it is making progress, it avoids "trap" sites for thaw, allowing it to learn cooling strategies that standard methods miss. See Table 9 for more information on these mechanisms.

## 5 DISCUSSION AND FUTURE WORK

We introduced BoreaRL, a multi-objective reinforcement learning framework for climate-adaptive boreal forest management. By integrating a physically-grounded simulator, it captures complex trade-offs between carbon sequestration and permafrost preservation. Our benchmarking highlights the challenge of asymmetric multi-objective optimization and demonstrates how adaptive episode selection helps avoid conflicting gradient "trap sites", yielding novel ecological insights into permafrost protection through appropriate management. BoreaRL offers a principled testbed for MORL and a high-impact platform for managing boreal carbon sinks to address climate change.

Despite these advances, several limitations constrain the current framework's scope. BoreaRL is designed as a *physically-grounded simulator* to enable controlled experimentation and generate *testable scientific hypotheses* (e.g., density-thaw relationships), not as a deployment-ready decision support system. Real-world deployment would require rigorous calibration against field data, expert validation, and integration with existing planning tools. Additionally, the long time horizons (50 years) and delayed rewards inherent in this domain create a challenging credit assignment problem that our benchmark isolates for study. While site selection demonstrates improved learning, MORL agents still struggle with the full complexity of environmental stochasticity, indicating that novel approaches are needed to fully realize the potential of RL-based climate-adaptive forest management.

Several concrete directions for future research emerge (see Appendix E):

1. **Multi-objective Extensions** incorporating economic (timber revenues, management costs) and biodiversity metrics.

2. **Advanced MORL Algorithms** utilizing meta-learning, non-linear scalarization, and continuous preference space optimization for asymmetric objectives.

3. **Real-world Validation and Deployment** using historical data and spatially explicit gridwise environments for continuous decision support.

## 6 ACKNOWLEDGEMENTS

We thank the Natural Sciences and Engineering Research Council of Canada (NSERC) for funding and supporting this research through the Alliance Mission grant - ALLRP 577126-2022 and the NSERC PDF program.

## 7 REPRODUCIBILITY STATEMENT

We provide all components needed to replicate our results. The main paper specifies the environment and simulator design (Table 1), the multi-objective formulation and baselines (Sections 3.2–3.4), and the experimental protocol with evaluation metrics used across figures. The Appendix documents the physics and implementation details of the simulator (Appendix B), the environment API and training paradigms (Appendix C), and the parameter sampling ranges and other configuration choices needed to recreate experiments; Source code associated with this project is attached as an anonymous supplementary zip, which includes the complete environment and simulator, training/evaluation scripts, configuration for both site-specific and generalist settings, and seeds for all reported runs. Figures in the paper reference the exact methods and settings compared so results can be matched to the corresponding configs and scripts.

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

## A    SUBMISSION DETAILS

### A.1    SOURCE CODE

Source code associated with this project can be found in the BoreaRL GitHub Repository.

### A.2    USE OF LARGE LANGUAGE MODELS

We used large language models (LLMs) in the following scoped, human-supervised ways: (i) Writing polish. Draft sections were refined for clarity, structure, and tone; all technical claims, numbers, and citations were authored and verified by us, and every LLM-suggested edit was line-reviewed to avoid introducing errors or unsupported statements. (ii) Retrieval & discovery. We used LLMs to craft and refine search queries to find related work and background resources; candidate papers were then screened manually, with citations checked against the original sources to prevent hallucinations. (iii) Research ideation. We used brainstorming prompts to surface alternative baselines, ablation angles, and failure modes; only ideas that survived feasibility checks and pilot experiments were adopted. (iv) Coding assistance (via Cursor). We used Cursor's inline completions and chat for boilerplate generation (tests, docstrings, refactors); all code was reviewed and benchmarked before inclusion. Across all uses, we ensured that LLM outputs never replaced human analysis, reproducibility artifacts, or empirical validation.

## B    FOREST SIMULATOR

### B.1    SIMULATOR PRINCIPLES AND FLOW

The BoreaRL benchmark is built upon a process-based simulator, *BoreaRL-Sim*, which is designed to be scientifically credible while remaining computationally efficient for reinforcement learning. The design adheres to several key principles:

- **Time-Scale Separation:** There is a clear separation between the agent's decision-making timescale and the physical simulation timescale. The RL agent performs a management action once per year. In response, *BoreaRL-Sim* resolves the energy, water, and carbon fluxes at a $n$-minute time-step (tunable) for the entire 365-day period, capturing the fine-grained diurnal and seasonal dynamics that govern the ecosystem.

- **Stochasticity and Robustness:** To ensure that learned policies are robust and not overfit to a single deterministic future, each $H$-year episode is driven by a unique, stochastically generated climate. At the start of an episode, a site latitude is sampled, which parametrically determines the mean climate characteristics (e.g., annual temperature, seasonality, growing season length). Then, daily weather (temperature, precipitation) is generated with stochastic noise, forcing the agent to learn strategies that are adaptive to climate variability.

- **Multi-Objective Core:** The simulator is fundamentally multi-objective. It tracks the two primary reward components: a carbon component that includes normalized net carbon change with stock bonuses and limit penalties, and an asymmetric thaw component derived from conductive heat flux to deep soil (permafrost proxy). These are returned as a reward vector $\boldsymbol{R}_t = [R_{carbon,t}, R_{thaw,t}]$ where $R_{thaw,t}$ penalizes warming more than it rewards cooling. This allows for the training of both specialist and generalist multi-objective agents.

- **Management-Physics Coupling:** The agent's actions (changing stand density and species composition) directly modulate the core physical parameters of the simulation, such as the Leaf Area Index (LAI), canopy albedo ($\alpha_{can}$), snow interception efficiency, and aerodynamic roughness. This creates a tight feedback loop where management decisions have physically-grounded consequences on the surface energy balance.

The simulation flow for a single RL step (one year) is orchestrated by the *ForestEnv* environment, which wraps the *BoreaRL-Sim* instance. The sequence is detailed in Appendix B.5.

## B.2 CORE PHYSICAL MODEL: ENERGY BALANCE

The core of *BoreaRL-Sim* is a multi-node thermodynamic model that solves the energy balance for the key components of the forest stand. The model uses standard, validated physical formulations found in major Land Surface Models like CLM5 (Fisher et al., 2019; Lawrence et al., 2019) and CLASSIC (Melton et al., 2020). For example, the state of the system is defined by the temperatures of five primary nodes: Canopy ($T_{can}$), Trunk ($T_{trunk}$), Snowpack ($T_{snow}$), Surface Soil ($T_{soil,surf}$), and Deep Soil ($T_{soil,deep}$). The temperature evolution of each node $i$ is governed by the net energy flux, following the principle:

$$C_i \frac{dT_i}{dt} = \sum F_{in,i} - \sum F_{out,i}$$

where $C_i$ is the heat capacity of the node and $F$ represents an energy flux in Watts per square meter ($Wm^{-2}$). The main flux equations for each node are detailed below.

### B.2.1 CANOPY NODE ($T_{can}$)

The canopy energy balance is the most complex, as it involves radiative, turbulent, and physiological fluxes. Unlike the soil and snow nodes which have significant thermal mass, the canopy has a relatively low heat capacity and equilibrates rapidly with the atmosphere. Therefore, we treat it as a diagnostic variable that reaches steady-state equilibrium at each timestep, rather than a prognostic variable with memory. Its temperature is solved iteratively to satisfy the condition where the net flux is zero:

$$0 = R_{net,can} - H_{can} - LE_{can} - G_{photo} - \Phi_{melt,can} - \Phi_{c,trunk}$$

The components are:

**$R_{net,can}$**: Net radiation, defined as the balance of incoming shortwave ($Q_{abs}$) and longwave ($L_{in}$) radiation against emitted longwave radiation ($L_{out}$). We use the Beer-Lambert Law for radiation extinction (Swinehart, 1962):

$$R_{net,can} = Q_{solar}(1 - \alpha_{can}) + \epsilon_{can}(L_{down,atm} + L_{up,ground}) - 2\epsilon_{can}\sigma T_{can}^4$$

**$H_{can}$**: Sensible heat flux, representing convective heat exchange with the air, governed by an aerodynamic conductance $h_{can}$:

$$H_{can} = h_{can}(T_{can} - T_{air})$$

**$LE_{can}$**: Latent heat flux from transpiration, modeled using the Priestley-Taylor formulation (Priestley & Taylor, 1972) modified by environmental stress factors for Vapor Pressure Deficit ($f_{VPD}$) and soil water content ($f_{SWC}$):

$$LE_{can} = \alpha_{PT} \frac{\Delta}{\Delta + \gamma} R_{net,can} \cdot f_{VPD} \cdot f_{SWC}$$

where $\alpha_{PT}$ is the Priestley-Taylor coefficient, $\Delta$ is the slope of the saturation vapor pressure curve, and $\gamma$ is the psychrometric constant.

**$G_{photo}$**: The energy sink used for photosynthesis. This is directly coupled to the carbon model via a light-use efficiency parameter ($LUE$), ensuring energy and carbon are conserved: $G_{photo} = GPP \cdot J_{per\_gC}$, where $J_{per\_gC}$ is the energy cost to fix one gram of carbon.

**$\Phi_{melt,can}$**: Energy sink for melting intercepted snow on the canopy, active only when $T_{can} \geq 273.15K$ and canopy snow exists.

**$\Phi_{c,trunk}$**: Conductive heat flux between the canopy and the tree trunks.

### B.2.2 TRUNK, SNOW, AND SOIL NODES

The energy balances for the ground-level nodes are primarily driven by their own radiation balance, turbulent exchange with the sub-canopy air, and conduction between adjacent nodes.

**Trunk Node ($T_{trunk}$):**

$$C_{trunk} \frac{dT_{trunk}}{dt} = H_{trunk} + \Phi_{c,can} + \Phi_{c,ground}$$

where $H_{trunk}$ is the sensible heat flux to the air, and $\Phi_{c,ground}$ represents conduction to either the snowpack or the surface soil, depending on snow cover.

**Snowpack Node ($T_{snow}$):**

$$C_{snow}\frac{dT_{snow}}{dt} = R_{net,snow} - H_{snow} - \Phi_{melt,snow} + \Phi_{c,soil} + \Phi_{c,trunk}$$

Here, $R_{net,snow}$ is the net radiation at the snow surface, which is high in albedo ($\alpha_{snow} \approx 0.8$). $\Phi_{melt,snow}$ is the energy sink for melting, active when the net energy flux is positive and $T_{snow} = 273.15K$. The thaw-degree-day reward component ($R_{thaw,t}$) is derived from the energy flux out of the deep soil layer into the permafrost boundary, providing a physically-based metric of permafrost degradation.

**Surface Soil Node ($T_{soil,surf}$):**

$$C_{soil,surf}\frac{dT_{soil,surf}}{dt} = R_{net,soil} - H_{soil} - LE_{soil} - \Phi_{c,deep} + \Phi_{c,snow} + \Phi_{c,trunk}$$

This balance is active when no snow is present. It includes latent heat of evaporation from the soil ($LE_{soil}$) and conductive flux to the deep soil layer ($\Phi_{c,deep}$).

**Deep Soil Node ($T_{soil,deep}$):**

$$C_{soil,deep}\frac{dT_{soil,deep}}{dt} = \Phi_{c,surf} - \Phi_{c,boundary}$$

The deep soil node integrates heat from the surface layer and loses heat to a fixed-temperature deep boundary, representing the top of the permafrost table. The permafrost thaw objective, $R_{thaw,t}$, is calculated as an asymmetric function of the cumulative annual positive and negative energy fluxes across this boundary, penalizing warming more than it rewards cooling.

## B.3 Key Simulator Sub-models

Layered on top of the core energy balance are modules for the water balance, carbon cycle, and stand dynamics. The simulator's realism is achieved through several key subsystems that work together to capture the complex dynamics of boreal forest ecosystems:

### B.3.1 Age-Structured Demography

The simulator implements an age-structured population model with five age classes: seedling (0-5 years), sapling (6-20 years), young (21-40 years), mature (41-100 years), and old (101+ years). Each age class has distinct canopy factors and light-use efficiency scaling, with natural transitions occurring annually. Thinning operations are preferentially applied to old trees to simulate realistic harvesting practices, while planting adds seedlings with the specified species mix. This age structure enables realistic representation of forest development trajectories and management constraints, as younger trees have different physiological properties and respond differently to environmental conditions.

### B.3.2 Advanced Weather Generation

The weather module generates latitude-dependent climate parameters including temperature-precipitation relationships that vary by season. Summer precipitation is positively correlated with temperature (reflecting convective rainfall patterns), while winter precipitation shows complex temperature dependencies (snow vs. rain thresholds). The system also models rain-induced suppression of diurnal temperature amplitude, where precipitation events dampen daily temperature swings through increased cloud cover and latent heat effects. This creates realistic weather patterns that influence forest dynamics through both direct physiological effects and indirect impacts on the energy balance.

### B.3.3 Disturbance Modeling

The simulator includes stochastic models for fire and insect outbreaks. Fire probability is conditioned on drought index (accumulated temperature and precipitation deficit), temperature thresholds

(fires more likely during hot periods), and species flammability (conifers more susceptible than deciduous). Insect outbreaks depend on winter temperature (warmer winters increase overwintering survival) and stand density (denser stands facilitate spread), with coniferous species being more susceptible to infestations. Both disturbances cause fractional mortality and route carbon appropriately: fire combusts biomass (releasing to atmosphere), while insect kill transfers dead biomass to soil carbon pools.

### B.3.4 HARVESTED WOOD PRODUCTS (HWP) ACCOUNTING

When thinning occurs, the simulator tracks carbon sequestration in harvested wood products rather than treating all removed biomass as immediate emissions. By default, most of the removed biomass carbon (typically 70-80%) is stored as HWP (contributing positively to the carbon objective), while a smaller percentage is lost during harvest operations (representing processing waste and immediate emissions). This creates an additional management incentive for sustainable harvesting practices and reflects the reality that forest products can provide long-term carbon storage.

### B.3.5 WATER BALANCE

The simulator tracks water in three main reservoirs: canopy-intercepted water, the snowpack on the ground, and soil water content (SWC).

**Snowpack ($SWE$):** Snow Water Equivalent on the ground ($SWE_{ground}$) and on the canopy ($SWE_{can}$) are tracked. The change in the ground snowpack is given by:

$$\Delta SWE_{ground} = P_{snow,throughfall} - M_{ground}$$

where $P_{snow,throughfall}$ is snow that passes through the canopy and $M_{ground}$ is melt, calculated from the energy balance. Canopy snow interception is a function of LAI and species type (conifers intercept more).

**Soil Water Content ($SWC$):** The soil is treated as a single bucket model. Its water content changes according to:

$$\Delta SWC = P_{rain,throughfall} + M_{can} + M_{ground} - ET - R_{off}$$

where inputs are rain and meltwater, and outputs are evapotranspiration ($ET$, coupled to the energy balance) and runoff ($R_{off}$), which occurs when $SWC$ exceeds its maximum capacity.

### B.3.6 CARBON CYCLE

The simulator tracks two primary carbon pools: living biomass ($C_{biomass}$) and soil organic carbon ($C_{soil}$). The net change in total carbon is a key reward component.

**Gross Primary Production (GPP):** Carbon uptake is calculated at each $n$-minute time-step using the Light Use Efficiency (LUE) model (Monteith, 1972), where GPP is proportional to absorbed photosynthetic radiation ($PAR_{abs}$) and is down-regulated by environmental stressors:

$$GPP = PAR_{abs} \cdot LUE \cdot f(VPD) \cdot f(SWC)$$

**Respiration ($R$):** Carbon losses occur via respiration from both biomass (autotrophic, $R_a$) and soil (heterotrophic, $R_h$). Both are modeled as a function of temperature using the standard Q10 response curve (Van't Hoff & Lehfeldt, 1900):

$$R = R_{base} \cdot Q_{10}^{((T-T_{ref})/10)}$$

where $R_{base}$ is a base respiration rate at a reference temperature $T_{ref}$.

**Pool Dynamics:** The biomass and soil carbon pools are updated annually based on the integrated fluxes. The net change in biomass is Net Primary Production (NPP = GPP - $R_a$) minus losses to litterfall and mortality. The net change in soil carbon is the sum of inputs from litterfall and mortality minus losses from heterotrophic respiration.

$$\Delta C_{biomass} = NPP - L_{fall} - M_{fire} - M_{insect} - M_{nat}$$
$$\Delta C_{soil} = L_{fall} + M_{deadwood} - R_h$$

### B.3.7 STAND DYNAMICS AND DISTURBANCES

The simulator includes modules for natural population changes and stochastic disturbances that impact forest structure and carbon.

**Natural Demography:** At the end of each year, the model calculates background mortality as a function of stand density (self-thinning) and a base stochastic rate. It also calculates natural recruitment (new seedlings), which is limited by available space.

**Fire Module:** A stochastic fire event can occur during the summer. The probability is conditioned on the stand's species composition (conifers are more flammable) and a running drought index. If a fire occurs, it causes fractional mortality and combusts a portion of the biomass carbon, releasing it from the system.

**Insect Module:** An annual check for an insect outbreak is performed. The probability is conditioned on the mean winter temperature (warmer winters increase survival) and stand density. An outbreak causes fractional mortality, primarily targeting coniferous species, with the dead biomass being transferred to the soil carbon pool.

### B.4 PERFORMANCE OPTIMIZATIONS

The simulator includes several performance optimizations for efficient computation:

**JIT Compilation:** The canopy energy balance solver uses Numba JIT compilation for significant speedup of computationally intensive components.

**Memory Management:** The environment implements automatic memory management for history tracking, preventing memory leaks during long training runs.

**Configurable Physics Backend:** The simulator supports two physics backends: a pure Python implementation for compatibility and a Numba JIT-compiled backend for improved performance during training.

### B.5 ANNUAL SIMULATION TIMELINE

The *BoreaRL-Sim* instance evolves over a one-year RL time step following a precise sequence of events. This ensures that management actions and natural processes occur in a logical order.

1. **Management Implementation:** At the beginning of the year (t=0), the agent's action is immediately implemented. Stems are added (planting) or removed (thinning) according to the action's density and species mix specifications. If thinning occurs, the corresponding carbon is removed from the $C_{biomass}$ pool, representing harvested timber. The age distribution of the forest is updated accordingly.

2. **Physical Parameter Update:** Based on the new stand structure (density, species mix, age), all physical parameters are recalculated. This includes LAI, canopy area, albedo, roughness length, and interception efficiencies. This step ensures the subsequent physical simulation uses properties that reflect the management action.

3. **Sub-Annual Physics Loop:** The simulator runs for 365 days, with time-steps per day depending on step resolution ($n$). In each time-step:

   - The weather forcing (temperature, radiation, precipitation) is updated based on the daily and diurnal cycles, with added stochastic noise.
   - The full energy, water, and carbon balance equations (see Appendix B.2 and B.3) are solved for the current state.
   - The temperatures of all nodes, snow water equivalent (SWE), and soil water content (SWC) are updated.
   - A check for a stochastic fire event is performed if conditions are met (summer, high drought index).
   - The dynamic carbon pools ($C_{biomass}$, $C_{soil}$) are updated based on the GPP and respiration fluxes of the time-step.

4. **End-of-Year Bookkeeping:** After the 365-day loop completes, several annual processes are resolved:

   - A stochastic check for an insect outbreak is performed.
   - Natural mortality (background and density-dependent) and recruitment are calculated and applied to the stand's age distribution.
   - All trees in the age distribution are aged by one year.
   - Final carbon pool values and the final stem density are calculated.

5. **Return Metrics:** The simulator calculates the final annual metrics needed for the RL reward. This includes the net change in total ecosystem carbon, positive and negative energy fluxes across the permafrost boundary, and various carbon pool states. These raw metrics are returned to the *ForestEnv* wrapper, which processes them into the normalized reward vector $\boldsymbol{R}_t = [R_{carbon,t}, R_{thaw,t}]$.

### B.6 PARAMETERIZATION

The *BoreaRL-Sim* is parameterized with a comprehensive set of physics-based parameters to ensure that trained agents are robust to environmental uncertainty and can generalize across diverse boreal forest conditions. In generalist mode, key parameters are sampled from uniform distributions at the start of each episode, while site-specific mode uses fixed parameter values. The parameterization spans climate forcing, soil properties, vegetation characteristics, and disturbance regimes, enabling realistic representation of boreal ecosystem variability.

The parameterization strategy ensures that trained agents encounter realistic environmental variability while maintaining physical consistency. Climate parameters are sampled from ranges representative of boreal forest latitudes, with temperature and precipitation patterns that capture the seasonal dynamics of northern ecosystems. Soil properties vary within ranges typical of boreal soils, including thermal conductivity and water-holding capacity. Vegetation parameters reflect the contrasting characteristics of coniferous and deciduous species, with different maximum leaf area indices and albedo values. Carbon cycle parameters are sampled from literature-based ranges for boreal ecosystems, ensuring realistic carbon fluxes and turnover rates. Disturbance parameters capture the stochastic nature of fire and insect outbreaks, with probabilities and impacts calibrated to boreal forest conditions. This comprehensive parameterization enables the environment to serve as a robust testbed for multi-objective forest management under climate uncertainty.

Table 4: Comprehensive parameter sampling ranges for the *BoreaRL-Sim*.

| Parameter | Description | Sampled Range |
|---|---|---|
| *Climate Forcing* | | |
| Latitude | Site latitude (°N) | [56.0, 65.0] |
| Mean Ann. Temp. Offset | Climate warming/cooling offset (°C) | [-10.0, -5.0] |
| Seasonal Amplitude | Seasonal temperature swing (°C) | [20.0, 25.0] |
| Diurnal Amplitude | Daily temperature variation (°C) | [4.0, 8.0] |
| Peak Diurnal Hour | Hour of maximum daily temperature | [3.0, 5.0] |
| Daily Noise Std | Temperature stochasticity (°C) | [1.0, 2.0] |
| Relative Humidity | Mean atmospheric humidity | [0.6, 0.8] |
| *Precipitation Patterns* | | |
| Summer Rain Prob | Daily summer precipitation probability | [0.10, 0.20] |
| Summer Rain Amount | Summer rainfall (mm/day) | [10.0, 20.0] |
| Winter Snow Prob | Daily winter snowfall probability | [0.15, 0.30] |

**Table 4 – continued from previous page**

| Parameter | Description | Sampled Range |
|---|---|---|
| Winter Snow Amount | Winter snowfall (mm/day) | [3.0, 8.0] |
| *Soil Properties* | | |
| Soil Conductivity | Thermal conductivity (W/m/K) | [0.8, 1.6] |
| Max Water Content | Soil water capacity (mm) | [100.0, 200.0] |
| Stress Threshold | Water stress threshold (fraction) | [0.3, 0.6] |
| Deep Boundary Temp | Permafrost boundary temperature (K) | [268.0, 272.0] |
| *Vegetation Characteristics* | | |
| Max LAI Conifer | Maximum leaf area index (conifers) | [3.0, 5.0] |
| Max LAI Deciduous | Maximum leaf area index (deciduous) | [4.0, 6.0] |
| Base Albedo Conifer | Canopy albedo (conifers) | [0.07, 0.11] |
| Base Albedo Deciduous | Canopy albedo (deciduous) | [0.15, 0.20] |
| *Carbon Cycle* | | |
| Base Respiration | Biomass respiration rate (kgC/m²/yr) | [0.30, 0.40] |
| Soil Respiration | Soil respiration rate (kgC/m²/yr) | [0.4, 0.6] |
| Q10 Factor | Temperature sensitivity of respiration | [1.8, 2.3] |
| Litterfall Fraction | Annual biomass turnover rate | [0.03, 0.04] |
| *Demography* | | |
| Natural Mortality | Annual mortality rate | [0.02, 0.03] |
| Recruitment Rate | Annual recruitment rate | [0.005, 0.015] |
| Max Natural Density | Maximum stand density (stems/ha) | [1500, 2000] |
| *Disturbances* | | |
| Fire Drought Threshold | Drought index for fire ignition | [20, 40] |
| Fire Base Probability | Annual fire probability | [0.0001, 0.0005] |
| Insect Base Probability | Annual insect outbreak probability | [0.02, 0.05] |
| Insect Mortality Rate | Mortality rate during outbreaks | [0.02, 0.05] |
| *Phenology* | | |
| Growth Start Day | Day of year for growth onset | [130, 150] |
| Fall Start Day | Day of year for senescence onset | [260, 280] |
| Growth Rate | Spring phenology rate | [0.08, 0.15] |
| Fall Rate | Autumn phenology rate | [0.08, 0.15] |

## C   MULTI-OBJECTIVE RL ENVIRONMENT

The BoreaRL environment implements a multi-objective reinforcement learning framework that wraps the physics simulator with standardized interfaces for training and evaluation. The environment conforms to the *mo-gymnasium* API standard and supports both site-specific and generalist training paradigms.

## C.1 ENVIRONMENT ARCHITECTURE

The environment consists of several key components:

**Observation Space:** The environment robust provides a rich observation vector that captures the current ecological state, historical information, and environmental context. The observation space varies between operational modes: generalist mode includes episode-level site parameters for robust policy learning, while site-specific mode uses a reduced observation space with fixed parameters for location-targeted optimization.

**Action Space:** The environment implements a discrete action space that encodes two management dimensions: stand density changes (thinning or planting) and species composition targets. Actions are encoded as single discrete values representing unique combinations of density change and conifer fraction targets, enabling efficient policy learning while maintaining interpretable management decisions.

**Reward Function:** The environment returns a 2-dimensional reward vector $[R_{carbon,t}, R_{thaw,t}]$ at each step. Both reward components are normalized to the range $[-1, 1]$ per step to ensure comparable scales for optimization.

- **Carbon Reward** ($R_{carbon,t}$)**:** Normalized by a factor of $2.0 \, \text{kg C m}^{-2} \, \text{yr}^{-1}$. It includes the net carbon change plus stock bonuses, with penalties for exceeding realistic carbon pools ($> 15 \, \text{kg C m}^{-2}$ biomass, $> 20 \, \text{kg C m}^{-2}$ soil).

- **Thaw Reward** ($R_{thaw,t}$)**:** Normalized by a factor of $40.0$ degree-days $\text{yr}^{-1}$. It is calculated as an asymmetric function of conductive heat flux to deep soil: $R_{thaw} \propto (\text{cooling flux}) - \alpha \times (\text{warming flux})$, where $\alpha = 2.5$ is a penalty factor that heavily penalizes warming.

Despite the comparable numerical ranges $[-1, 1]$, the thaw objective is significantly harder to optimize due to this asymmetric penalty $\alpha$ and the conflicting physics of the domain (e.g., snow insulation vs. albedo effects), rather than a difference in reward magnitude. A theoretical optimal return for both objectives over a 50-year episode is estimated at approximately $50.0$.

**Episode Structure:** Each episode consists of 50 annual management decisions, with each decision followed by a full 365-day physical simulation.

## C.2 TRAINING PARADIGMS

The environment supports two distinct training paradigms:

**Site-Specific Mode:** Designed for controlled studies and location-targeted optimization, this mode uses deterministic weather patterns, fixed site parameters, and reduced observation dimensionality. The environment uses a fixed weather seed, zero temperature noise, and deterministic initial conditions, providing reproducible results for systematic ablation studies.

**Generalist Mode:** Designed for robust policy learning under environmental stochasticity, this mode samples unique weather sequences and site parameters for each episode. The environment includes episode-level site parameters in the observation space, enabling policies to adapt to diverse forest conditions and climate variability.

## C.3 PREFERENCE CONDITIONING

The environment supports preference-conditioned training through a preference weight input in the observation space. This enables training single policies that can adapt to different objective weightings without retraining. Both fixed preference training (for controlled studies) and randomized preference sampling (for robust generalist policies) are supported.

## C.4 RL HYPERPARAMETERS

We evaluate three multi-objective RL algorithms across different training paradigms. The agents were trained using the hyperparameters listed in Table 5.

Table 5: Hyperparameters for Multi-Objective RL Agent Training

| Hyperparameter | Variable $\lambda$ EUPG | PPO Gated | Curriculum PPO |
|---|---|---|---|
| Framework | morl-baselines | Custom PyTorch | Custom PyTorch |
| Learning Rate | $1 \times 10^{-3}$ | $3 \times 10^{-4}$ | $3 \times 10^{-4}$ |
| Discount Factor ($\gamma$) | 1.0 | 0.99 | 0.99 |
| Network Architecture | [128, 64] | [64, 64] | [64, 64] |
| GAE Lambda | N/A | 0.95 | 0.95 |
| Clip Coefficient | N/A | 0.2 | 0.2 |
| Rollout Steps | N/A | 2048 | 2048 |
| Batch Size | N/A | 64 | 64 |
| Update Epochs | N/A | 10 | 10 |
| Curriculum Threshold | N/A | N/A | 0.5 |
| Plant Gate | N/A | Enabled | Enabled |
| Total Timesteps (Generalist) | $3 \times 10^5$ | $3 \times 10^5$ | $3 \times 10^5$ |
| Total Timesteps (Site-specific) | $1 \times 10^5$ | $1 \times 10^5$ | $1 \times 10^5$ |

## C.5 CURRICULUM PPO MECHANISM

The Curriculum PPO agent employs an adaptive episode selection mechanism to stabilize learning in the generalist setting. The mechanism consists of two components:

1. **Episode Selector Network:** A fixed, randomly initialized neural network $f_\phi : \mathcal{O}_{site} \rightarrow [0, 1]$ that projects site features to a scalar score. This network is not trained via gradient descent but provides a consistent hashing of the site space. We employ a fixed projection to establish a minimal baseline for curriculum efficacy, demonstrating that the adaptive thresholding mechanism itself is sufficient for stabilization without the added complexity of learning a site-value function. While an optimal ordering of the curriculum could potentially yield further improvements, our results show that even this random ordering provides significant benefits.

2. **Adaptive Threshold:** A dynamic threshold $\tau$ that determines whether to accept an episode for training ($f_\phi(s) > \tau$). The threshold is updated based on the relative performance of selected versus skipped episodes. If the agent performs better on selected episodes (indicating mastery of the current subset), the threshold is decreased ($\tau \leftarrow \tau \times 0.999$) to *expand* the training distribution. If performance on selected episodes is worse than skipped ones, the threshold is increased ($\tau \leftarrow \tau \times 1.001$) to *contract* the curriculum to a smaller, more manageable subset.

This approach creates an automatic "breathing" curriculum that expands and contracts the effective training distribution based on the agent's current competence.

## C.6 OBSERVATION SPACE DETAILS

The observation space structure varies between operational modes. In generalist mode, the observation vector contains 105 dimensions as detailed in Table 6, while site-specific mode uses a reduced observation space of 43 dimensions that excludes variable site parameters. The observation vector is designed to provide information about the current ecological state, historical trends, and environmental context to enable effective policy learning.

Table 6: Detailed breakdown of the observation space structure (generalist mode).

| Index | Description | Normalization |
|---|---|---|
| *Category 1: Preference Input (1 dimension)* | | |

**Table 6 – continued from previous page**

| Index | Description | Normalization |
|-------|-------------|---------------|
| 0 | Carbon Preference Weight ($w_C$) | $[0, 1]$ (no change) |
| *Category 2: Current Ecological State (4 dimensions)* | | |
| 1 | Year | $year/50$ |
| 2 | Stem Density (stems ha$^{-1}$) | $density/1500$ |
| 3 | Conifer Fraction | $[0, 1]$ (no change) |
| 4 | Total Carbon Stock (kgC m$^{-2}$) | $(biomass_C + soil_C)/50$ |
| *Category 3: Site Climate Parameters (6 dimensions)* | | |
| 5 | Latitude (°N) | $(lat - 50)/20$ |
| 6 | Mean Annual Temperature (°C) | $(T_{mean} + 10)/20$ |
| 7 | Seasonal Temperature Amplitude (°C) | $T_{amp}/30$ |
| 8 | Growth Start Day (DOY) | $growth\_day/365$ |
| 9 | Fall Start Day (DOY) | $fall\_day/365$ |
| 10 | Growing Season Length (days) | $(fall\_day - growth\_day)/200$ |
| *Category 4: Disturbance History (6 dimensions)* | | |
| 11-12 | Fire Mortality Fraction (last 2yr) | $[0, 1]$ (fraction) |
| 13-14 | Insect Mortality Fraction (last 2yr) | $[0, 1]$ (fraction) |
| 15-16 | Drought Index (last 2yr) | $index/100$ |
| *Category 5: Carbon Cycle Details (7 dimensions)* | | |
| 17 | Recent Biomass C Change | $(change + 0.5)/1.0$ |
| 18 | Recent Soil C Change | $(change + 0.2)/0.4$ |
| 19 | Recent Total C Change | $(change + 0.7)/1.4$ |
| 20 | Recent Natural Mortality | $mortality/0.5$ |
| 21 | Recent Litterfall | $litterfall/2.0$ |
| 22 | Recent Thinning Loss | $(loss + 0.5)/1.0$ |
| 23 | Recent HWP Stored | $hwp/0.5$ |
| *Category 6: Management History (4 dimensions)* | | |
| 24 | Recent Density Action Index | $action/4$ |
| 25 | Recent Mix Action Index | $action/4$ |
| 26 | Recent Density Change | $(change + 100)/200$ |
| 27 | Recent Mix Change | $change$ (no change) |
| *Category 7: Age Distribution (10 dimensions)* | | |
| 28-32 | Conifer Age Fractions (5 classes) | $[0, 1]$ (fraction) |
| 33-37 | Deciduous Age Fractions (5 classes) | $[0, 1]$ (fraction) |
| *Category 8: Carbon Stocks (2 dimensions)* | | |
| 38 | Normalized Biomass Stock | $biomass/50$ |
| 39 | Normalized Soil Stock | $soil/50$ |
| *Category 9: Penalty Information (3 dimensions)* | | |
| 40 | Biomass Limit Penalty | $penalty/0.5$ |
| 41 | Soil Limit Penalty | $penalty/0.5$ |
| 42 | Max Density Penalty | $penalty/1.0$ |
| *Category 10: Site Parameter Context (62 dimensions, generalist only)* | | |
| 43-104 | Site-specific physics parameters | Normalized to $[0, 1]$ ranges |

The normalization strategy ensures that all components are scaled to approximately $[0, 1]$ ranges for stable learning, while preserving the relative magnitudes and relationships between different ecological variables. The preference input enables preference-conditioned policies to adapt their behavior based on the desired objective weighting.

## C.7 ACTION SPACE ENCODING DETAILS

The environment uses a discrete action space with 25 unique actions (5 density actions × 5 conifer fractions) as detailed in Table 7. Actions are encoded as single integer values where the density action index and conifer fraction index are combined using integer division and modulo operations.

Table 7: Action Space Encoding for Forest Management

| Action Index | Density Change (stems/ha) | Conifer Fraction |
|---|---|---|
| 0-4 | -100 | 0.0, 0.25, 0.5, 0.75, 1.0 |
| 5-9 | -50 | 0.0, 0.25, 0.5, 0.75, 1.0 |
| 10-14 | 0 | 0.0, 0.25, 0.5, 0.75, 1.0 |
| 15-19 | +50 | 0.0, 0.25, 0.5, 0.75, 1.0 |
| 20-24 | +100 | 0.0, 0.25, 0.5, 0.75, 1.0 |

Management constraints include thinning restrictions to maintain a minimum density of 150 stems/ha, with thinning operations removing oldest trees first (101+ years) when available. Planting operations add seedlings (0-5 years) to the stand, with a maximum density of 2000 stems/ha that cannot be exceeded. If planting is attempted at maximum density, a penalty is applied. Species composition is controlled by the conifer fraction parameter, allowing for mixed-species management strategies.

**Carbon Accounting:** Net carbon change calculations include biomass, soil, and HWP carbon components. Carbon limits are enforced with penalties rather than hard caps, with biomass carbon limited to 15.0 kg C/m² and soil carbon limited to 20.0 kg C/m². Excess carbon beyond these limits incurs proportional penalties to discourage unrealistic carbon accumulation.

**Age Distribution Management:** The system tracks five age classes: seedling (0-5), sapling (6-20), young (21-50), mature (51-100), and old (101+ years). Natural mortality and recruitment occur annually, while management actions modify the age distribution based on species preferences. Age-weighted canopy factors affect light use efficiency and growth rates, creating realistic stand dynamics.

## C.8 REWARD FUNCTION MATHEMATICAL FORMULATION

The reward function returns a two-dimensional vector $[r_{carbon}, r_{thaw}]$ with the following components:

### C.8.1 CARBON REWARD ($r_c$)

$$r_c = \text{clip}(c_n + s_b + h_b - p_l - p_d - p_i, -1.0, 1.0)$$

Where:

$$c_n = \text{clip}(\frac{\Delta C}{2.0}, -1.0, 1.0)$$

$$s_b = 0.0 \times \text{clip}(\frac{C_t}{50.0}, 0.0, 1.0)$$

$$h_b = 0.0 \times \text{clip}(\frac{h}{1.0}, 0.0, 1.0)$$

$$p_l = p_b + p_s$$

$$p_b = \frac{e_b}{15.0} \times 0.5$$

$$p_s = \frac{e_s}{20.0} \times 0.5$$

$$p_d = \begin{cases} 1.0 & \text{if } d \geq 2000 \\ 0.0 & \text{otherwise} \end{cases}$$

### C.8.2 THAW REWARD ($r_t$)

$$r_t = \text{clip}(\frac{a_t}{40.0}, -1.0, 1.0)$$

Where:

$$a_t = f_n - 2.5 \times f_p$$

The carbon reward $r_c$ consists of normalized net carbon change $c_n$ (where $\Delta C$ is the net carbon change including HWP in kg C/m²/yr), stock bonus $s_b$ is based on total carbon stock $C_t$, and HWP sales bonus $h_b$ is based on HWP carbon stored $h$. Penalties include total limit penalties $p_l$ (sum of biomass penalty $p_b$ and soil penalty $p_s$, where $e_b$ and $e_s$ are carbon excesses beyond limits), density penalty $p_d$ (applied when stem density $d$ exceeds 2000 stems/ha), and ineffective action penalties $p_i$. The thaw reward $r_t$ is based on asymmetric thaw $a_t$ (degree-days/yr), which combines positive heat flux $f_p$ and negative heat flux $f_n$ to deep soil (permafrost proxy) with a 2.5:1 penalty ratio for warming versus cooling.

Regarding validation, both carbon and thaw rewards are constructed using existing common knowledge from literature about how these fluxes operate. Growth of carbon stock, carbon capacity of forest and soil, overplanting, excessive thinning, etc are common ways to think about the health of a forest in forest management. Thaw degree days and fluxes into and out of the soil are common ways to calculate permafrost thaw. Apart from this existing base, more components can be added to these depending on user preference and is subjective. Therefore, in some sense, these reward formulations are already validated from existing literature.

### C.9 COMPUTATIONAL COMPLEXITY AND RUNTIME ANALYSIS

Computational cost per episode is ~12-15 seconds per 50-year episode on a standard CPU (Intel i7/i9 or AMD Ryzen, single core). With Numba JIT it takes about ~5-7 seconds. The variance depends on the number of disturbances (fire/insect events trigger additional computation) and whether the canopy energy balance solver converges quickly (dependent on weather conditions).

In *Generalist Mode*, the total timesteps is 300,000 (6,000 episodes × 50 steps/episode) and training time is ~8-12 hours on a standard CPU workstation. In *Site-Specific Mode*, the total timesteps is 100,000 (2,000 episodes × 50 steps/episode) and training time is ~3-4 hours on a standard CPU workstation. These estimates include forward simulation (physics + reward computation), PPO policy/value network updates, Logging and checkpointing.

The primary computational cost is the *sub-daily physics loop*. For each of the 50 annual timesteps, the simulator runs 365 days × (1440 minutes / 30-minute resolution) = 17,520 physics steps. The canopy energy balance solver (iterative Newton-Raphson) accounts for ~60-70% of this cost. Training throughput scales well with CPU cores. With 16 parallel workers, generalist training can be completed in ~1-2 hours.

Per-environment memory footprint is ~50-100 MB. Training a single PPO agent (network parameters, replay buffer) is ~200-500 MB. Total for 16 parallel envs + agent is ~2-3 GB RAM, easily feasible on modern workstations.

In absolute terms, BoreaRL is trainable on commodity hardware (laptop or workstation). A full training run costs $1 in cloud compute (AWS EC2 c5.4xlarge). Relative to other physically-grounded simulators, BoreaRL is efficient, achieving a balance between physical realism and RL tractability. We will be exploring JAX/GPU acceleration in future versions.

## D ADDITIONAL RESULTS

This section provides additional analysis supporting the claims in the main paper.

### D.1 SITE INFLUENCE ON THAW PERFORMANCE

Site characteristics strongly influence thaw performance and learning stability. Certain sites enable much higher thaw objective values than others, demonstrating that site selection fundamentally determines achievable performance regardless of management decisions. The high volatility in thaw

reward learning across algorithms supports the importance of curriculum-based approaches for stable learning.

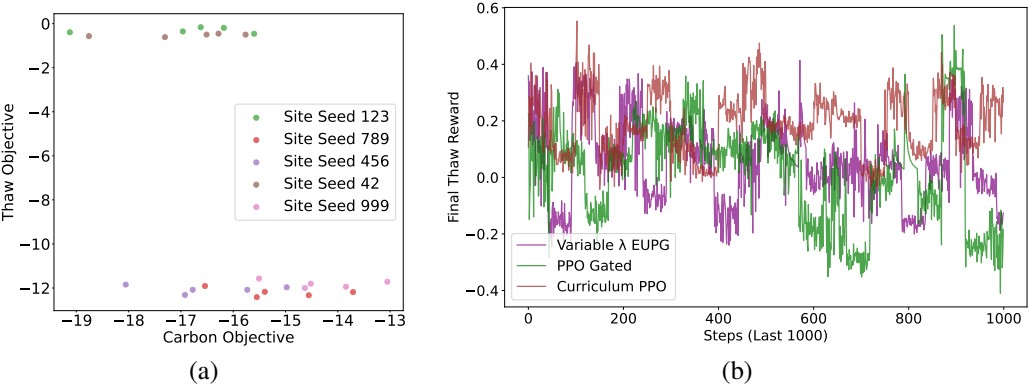

Figure 5: Site influence on thaw performance. (a) Thaw objective clustering by site characteristics. (b) Training volatility in thaw reward learning across algorithms.

## D.2 EMPIRICAL TRADE-OFF COVERAGE OF METHODS

Figure 6 shows the carbon-thaw trade-offs achieved by different algorithms.

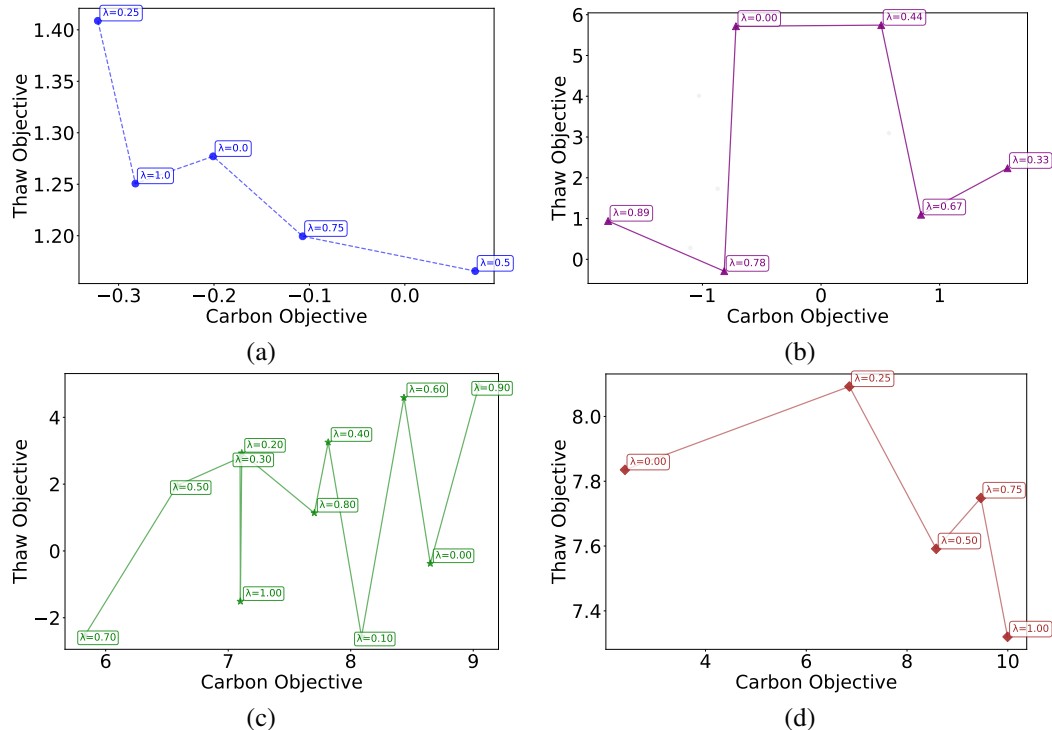

Figure 6: Empirical trade-off coverage for different algorithms. We plot for all weights (averaged across 100 episodes each; summed over 50 steps) to visualize the density and range of learned policies. The rewards are (a) Fixed-Weight Composition (b) Variable $\lambda$ EUPG (c) PPO Gated (d) Curriculum PPO.

**Lambda Monotonicity Analysis:** Curriculum PPO achieves the best preference control with 50% monotonicity violations, while other methods show poor preference adherence: Fixed-Weight

(75%), Variable $\lambda$ EUPG (66.7%), and PPO Gated (100% violations). Curriculum PPO provides the most reliable trade-off behavior for practical applications.

### D.3  MECHANISTIC EVIDENCE AND ANALYSIS

This section provides detailed mechanistic evidence supporting the claims made in the main text, linking agent behaviors to physical processes.

Table 8: Correlation between Growing Season Length and Thaw Protection Mechanisms. Longer growing seasons are strongly correlated with better thaw protection, driven by increased transpiration cooling and reduced ground radiation.

| Variable | Correlation ($r$) | Interpretation |
|---|---|---|
| Thaw Reward | +0.65 | Longer season $\rightarrow$ Better Thaw protection |
| Latent Heat (LE) | +0.82 | Longer season $\rightarrow$ More transpiration cooling |
| Ground Radiation | -0.75 | Longer season $\rightarrow$ Less solar heat on soil |

Table 9: Causal Chain of Learned Strategies. Different algorithms converge to distinct local optima with specific physical mechanisms, outcomes, and algorithmic causes.

| Agent | Action Strategy | Physical Mechanism | Outcome | Algorithmic Cause & Evidence |
|---|---|---|---|---|
| PPO Gated | Carbon Farming (High Density, High Conifer) | High Biomass: Maximizes Carbon. Low Albedo: Conifers absorb heat. | High Carbon ($\approx 9.0$) High Warming ($\approx -5.0$) | Gradient Dominance: Dense Carbon signal overpowers noisy Thaw signal. *Evid:* 100% $\lambda$-violations; Thaw $\approx -5.0$. |
| Variable $\lambda$ | Thaw Avoidance (Low Density, Deciduous) | High Albedo: Reflects sunlight. Low Biomass: Minimizes Carbon. | Low Carbon ($\approx 1.0$) Max Cooling ($\approx 8.0$) | Policy Collapse: Conflicting gradients lead to risk-averse "inaction". *Evid:* Lowest Reward (1.7); Hypervol 14.2. |
| Curriculum | Balanced / Cooling (Mod. Density, Mixed) | High Transpiration: Cools air. Mod. Albedo: Mixed reflection. | Good Carbon ($\approx 8.0$) Good Cooling ($\approx 6.0$) | Gradient Filtering: Removes "trap" sites, enabling complex learning. *Evid:* High Hypervol (84.3); Sparsity 0.12. |

Table 10: Comparison of Thaw Reward Formulations and Agent Behavior. The Asymmetric formulation forces strong avoidance of warming, whereas symmetric formulations allow small warming trade-offs.

| Formulation | Warming Penalty | Agent Behavior | Resulting Warming Flux |
|---|---|---|---|
| Raw DD | Symmetric (1.0) | Accepts small warming | $\approx 5.0$ MJ |
| Contrast | Ratio-based | Accepts warming if Cooling high | $\approx 2.0$ MJ |
| Asymmetric | Strong (2.5x) | Avoids Strongly | $\approx 0.1$ MJ |

Table 11: Physical Conflicts between Carbon and Permafrost Objectives.

| Physical Variable | Effect on Carbon ($C$) | Effect on Permafrost ($T$) | Conflict? |
|---|---|---|---|
| Stem Density | ↑ (More Biomass) | ↓ (Interception) / ↓ (Absorption) | Yes (Complex) |
| Conifer Fraction | ↑ (Higher Density) | ↓ (Lower Albedo → Warmer) | Yes (Direct) |
| LAI | ↑ (More Growth) | ↑ (Shading/Cooling) | No (Synergy) |

Table 12: Curriculum Selection Statistics: Characteristics of Accepted vs. Rejected Sites. The curriculum filters out sites with high warming potential.

| Site Category | Avg. Latitude | Avg. Potential Warming Flux |
|---|---|---|
| Accepted Sites | High ($> 60°$N) | Low ($\approx 0.5$ MJ) |
| Rejected Sites | Low ($< 60°$N) | High ($\approx 8.0$ MJ) |

### D.4 DETAILED OBJECTIVE ANALYSIS

Table 13 provides a detailed breakdown of the Carbon and Thaw objectives achieved by RL baselines. Curriculum PPO baseline consistently achieves high values in both objectives, demonstrating superior trade-off management. In contrast, PPO Gated achieves high Carbon scores but suffers significant penalties in the Thaw objective, often resulting in negative values due to the asymmetric warming penalty. Variable $\lambda$ EUPG fails to learn effective strategies for either objective, clustering near zero.

Table 13: Detailed breakdown of Carbon and Thaw objectives for RL baselines. Values correspond to means and errors represent standard deviation over 100 evaluation episodes.

| Method & Preference ($\lambda$) | Carbon Objective | Thaw Objective |
|---|---|---|
| **Curriculum PPO** | | |
| $\lambda = 0.00$ | $2.5 \pm 1.5$ | $7.8 \pm 1.2$ |
| $\lambda = 0.25$ | $7.0 \pm 1.8$ | $8.1 \pm 1.4$ |
| $\lambda = 0.50$ | $8.5 \pm 2.0$ | $7.6 \pm 1.5$ |
| $\lambda = 0.75$ | $9.5 \pm 1.5$ | $7.7 \pm 1.3$ |
| $\lambda = 1.00$ | $10.0 \pm 1.0$ | $7.3 \pm 1.6$ |
| **PPO Gated** | | |
| $\lambda = 0.00$ | $8.8 \pm 2.0$ | $-0.2 \pm 2.5$ |
| $\lambda = 0.20$ | $7.2 \pm 2.0$ | $2.5 \pm 2.5$ |
| $\lambda = 0.40$ | $7.7 \pm 1.5$ | $3.1 \pm 2.0$ |
| $\lambda = 0.60$ | $8.4 \pm 1.5$ | $4.2 \pm 2.0$ |
| $\lambda = 0.80$ | $7.6 \pm 1.5$ | $1.0 \pm 2.0$ |
| $\lambda = 1.00$ | $7.1 \pm 1.0$ | $-1.5 \pm 1.5$ |
| **Variable $\lambda$ EUPG** | | |
| $\lambda = 0.00$ | $-0.7 \pm 1.5$ | $5.8 \pm 2.0$ |
| $\lambda = 0.33$ | $1.7 \pm 1.5$ | $2.2 \pm 2.0$ |
| $\lambda = 0.44$ | $0.5 \pm 1.5$ | $5.8 \pm 2.0$ |
| $\lambda = 0.67$ | $0.9 \pm 1.5$ | $1.1 \pm 1.5$ |
| $\lambda = 0.78$ | $-0.8 \pm 1.5$ | $-0.2 \pm 1.5$ |
| $\lambda = 0.89$ | $-1.8 \pm 1.5$ | $1.0 \pm 1.5$ |

### D.5 TRAINING DYNAMICS AND STRATEGY EVOLUTION

Figure 7 shows training dynamics for conifer fraction and stem density, revealing distinct algorithmic learning patterns. PPO Gated pursues aggressive early carbon strategies with high conifer fractions and rapid density growth followed by decline. Curriculum PPO shows steady learning with moderate improvements in both metrics. Variable $\lambda$ EUPG maintains conservative species composition but exhibits sustained density growth throughout training. PPO Gated shows strong generalization from training to evaluation (see Fig. 4a,b), while Variable $\lambda$ EUPG exhibits potential overfitting with poor generalization in density management. Curriculum PPO demonstrates stable learning across both phases.

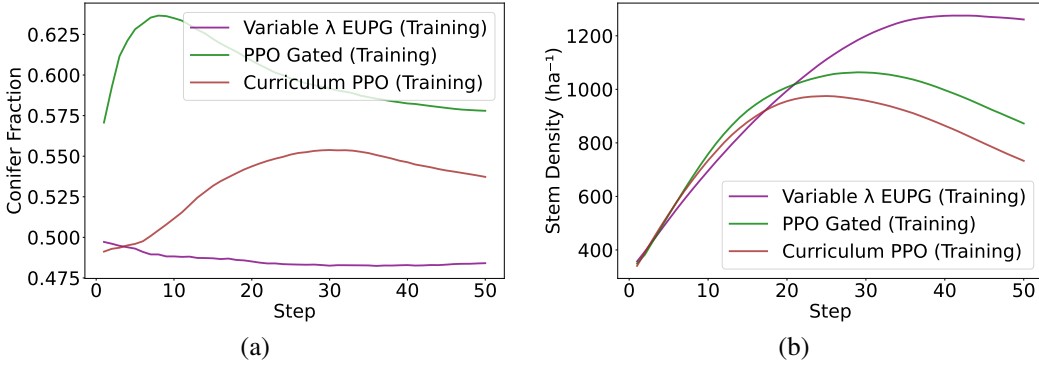

(a)                                                      (b)

Figure 7: Training dynamics of algorithm strategies. (a) Conifer fraction evolution during training. (b) Stem density evolution during training.

### D.6 MULTI-OBJECTIVE TRADE-OFFS AND CLIMATE ADAPTATION STRATEGIES

Figure 8 shows algorithm performance under varying environmental conditions and objective trade-offs. Curriculum PPO achieves superior multi-objective performance with high rewards in both carbon and thaw objectives. PPO Gated prioritizes carbon performance, while Variable $\lambda$ EUPG shows inconsistent exploration.

### D.7 INDIVIDUAL OBJECTIVE PERFORMANCE ANALYSIS

Figure 9 shows individual carbon and thaw objective performance for each algorithm. PPO Gated and Curriculum PPO achieve superior carbon performance, while Curriculum PPO dominates thaw optimization. Variable $\lambda$ EUPG shows poor performance in both objectives. Curriculum PPO's balanced multi-objective approach explains its superior scalarized performance.

### D.8 FOREST DEMOGRAPHICS AND COMPOSITIONAL DYNAMICS

This section provides an analysis of forest demographic trajectories and compositional preferences, offering detailed insights into how different algorithms manage forest structure and species composition over time. Figure 10 presents two complementary analyses: age-class specific stem density trajectories and conifer fraction distributions during training and evaluation.

**Age-Class Trajectory Analysis:** The age-class trajectories (Panels a-e) reveal distinct management strategies across different forest developmental stages. For younger age classes (seedling, sapling, young), all algorithms show initial high stem densities followed by rapid decline, reflecting natural mortality and competition processes. However, the recovery patterns differ: Variable $\lambda$ EUPG demonstrates sustained high densities in mature and old age classes, particularly for deciduous trees, suggesting a strategy focused on long-term forest productivity and carbon storage. PPO Gated shows strong early growth in conifer age classes but experiences more pronounced declines, indicating a strategy that prioritizes rapid establishment but may sacrifice long-term stability. Curriculum PPO exhibits intermediate patterns, balancing growth and sustainability across age classes.

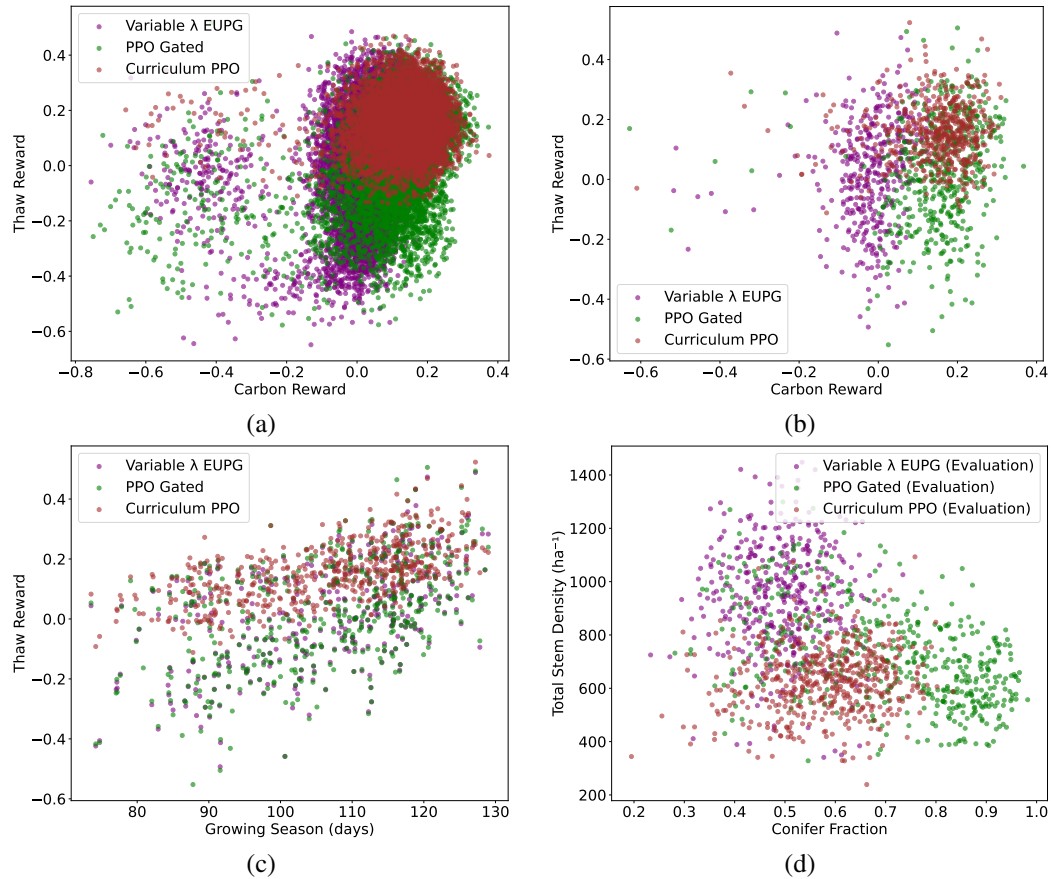

Figure 8: Algorithm performance analysis. (a) Carbon vs. thaw rewards during training. (b) Carbon vs. thaw rewards during evaluation. (c) Growing season vs. thaw reward during evaluation. (d) Forest demographics during evaluation.

These trajectory differences explain the observed carbon and thaw performance variations, as mature and old forests contribute significantly to both carbon sequestration and permafrost protection through their insulating effects.

**Compositional Strategy Analysis:** The conifer fraction distributions (Panels f-g) reveal fundamental differences in species composition preferences. During training (Panel f), Variable $\lambda$ EUPG shows a narrow, peaked distribution around 0.45-0.5 conifer fraction, indicating a lack of specialized strategy. PPO Gated exhibits a broader distribution with preference for higher conifer fractions (0.6-0.9), suggesting a strategy that promotes conifer dominance. Curriculum PPO shows intermediate preferences, with a broader distribution centered around 0.55-0.6, indicating adaptive compositional management. The evaluation distributions (Panel g) confirm these training preferences, with Variable $\lambda$ EUPG maintaining moderate conifer fractions (0.35-0.65), PPO Gated strongly favoring high conifer fractions (0.75-0.95), and Curriculum PPO showing balanced preferences around 0.6-0.65. These compositional strategies directly impact both carbon and thaw objectives: higher conifer fractions generally support carbon sequestration through increased biomass, while balanced compositions may better support permafrost protection through modified energy and water fluxes.

# E    EXTENDED FUTURE WORK

**Additional Multi-Objective Approaches:** Future work will incorporate evolutionary algorithms (e.g., NSGA-II, MOEA/D) as population-based alternatives, hypervolume-based methods to directly

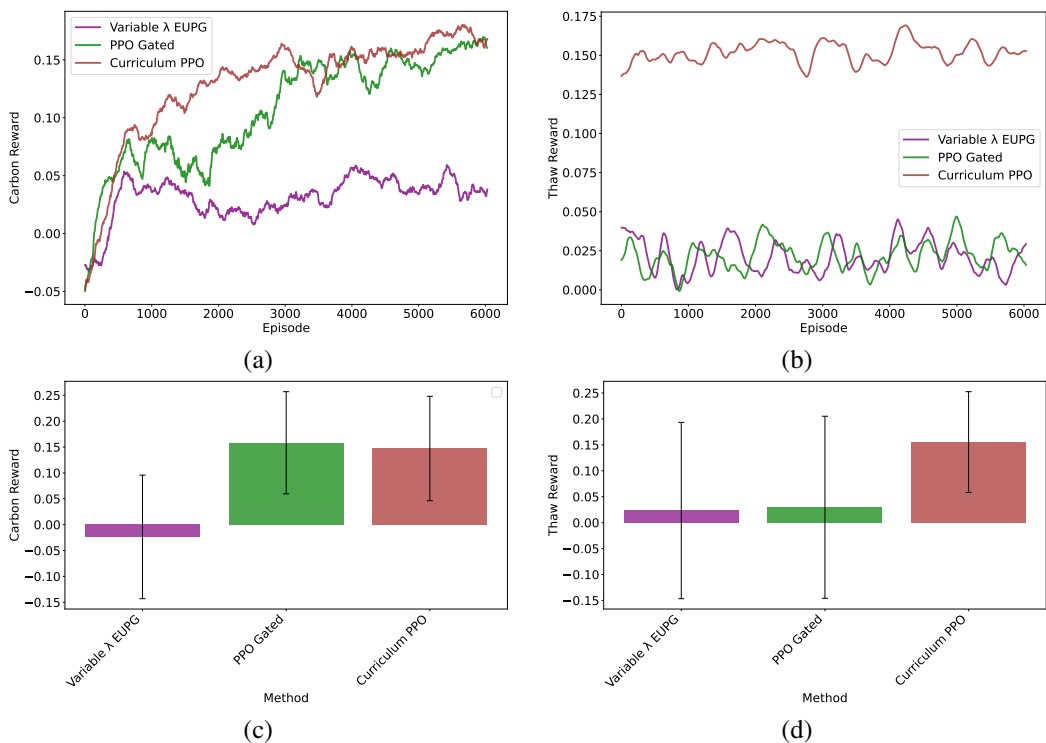

Figure 9: Individual objective performance analysis. (a) Carbon learning curves during training. (b) Thaw learning curves during training. (c) Carbon performance during evaluation. (d) Thaw performance during evaluation.

optimize trade-off quality, and model-based MORL to improve efficiency in long-horizon permafrost dynamics.

**Spatial and Temporal Scaling:** Develop hierarchical action spaces for landscape-scale management, incorporating spatial interactions between stands, temporal coordination of management activities, and integration with regional climate models for improved environmental forecasting.

**Computational Acceleration:** Enable massive parallelization on GPUs/TPUs with JAX/PyTorch. This would allow for end-to-end vectorization, significantly faster training times, and the ability to scale to much larger populations of agents or more complex environmental simulations.

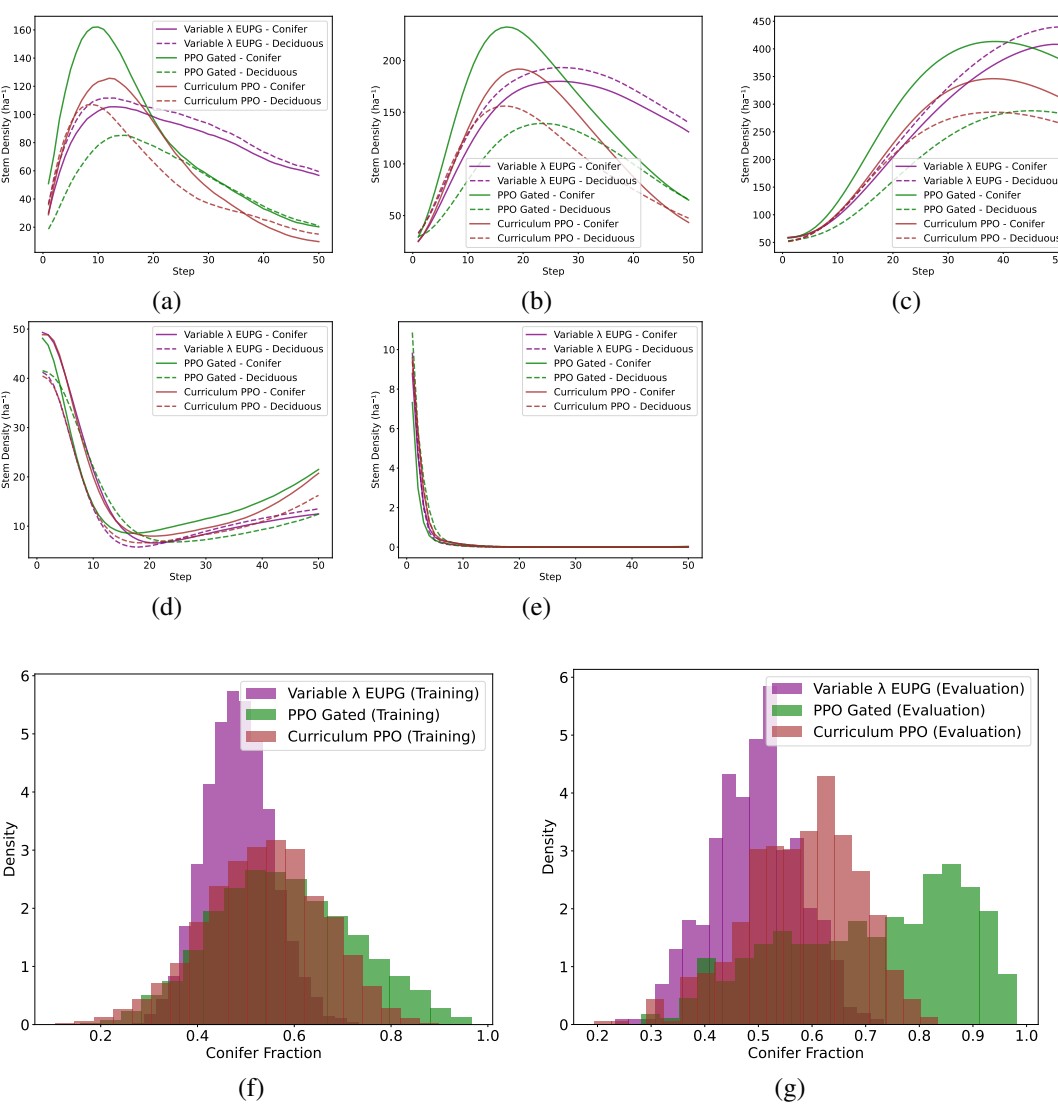

Figure 10: Detailed forest demographics and compositional analysis. (a) Seedling trajectory showing stem density evolution over 50 simulation steps. (b) Sapling trajectory illustrating early forest development patterns. (c) Young trajectory demonstrating intermediate growth dynamics. (d) Mature trajectory revealing long-term forest structure management. (e) Old trajectory showing late-stage forest dynamics. (f) Conifer fraction distribution during training phase, revealing compositional exploration strategies. (g) Conifer fraction distribution during evaluation phase, showing learned compositional preferences. All panels illustrate how different algorithms (Variable $\lambda$ EUPG in purple, PPO Gated in green, Curriculum PPO in brown) manage forest structure and species composition across developmental stages.

