# OpenReview forum: "BoreaRL: A Multi-Objective Reinforcement Learning Environment for Climate-Adaptive Boreal Forest Management"
_ICLR.cc/2026/Conference — ICLR 2026 Poster_

### Official Review · Reviewer_3zMh · 2025-10-24

**Soundness:** 2
**Presentation:** 3
**Contribution:** 3
**Rating:** 4
**Confidence:** 3

**Summary:**

The authors propose a novel RL environment modelling the management of boreal forests. The environment tasks RL algorithms with two objectives: maximizing carbon sequestration and  minimizing permafrost thaw. The environment is built on a complex thermodynamic simulator that models the energy balance of different parts of the forest stand. The simulator also models the carbon cycle, water balance and various weather patterns. Finally, the environment allows for training only on specific site or from a site/weather sampled from some distribution.

To test their environment, the authors test both expected utility policy gradient (with different preference weights) and PPO. Experimentally, they observe that it is easier to optimize for carbon objectives than thaw objectives. To address the difficulty, they use curriculum learning and show that it outperforms other methods and learns to improve performance on thawing objectives.

**Strengths:**

- The paper is very well-written and motivated, it does a good job of presenting the subject as a valuable application area for ML.
- The work is novel and appears to address a gap in the literature.
- The environment involves a significant amount of work and attention to detail to model a large amount of aspects of the real world in the physical simulation.
- The authors test various RL methods on the environment and carefully analyze their performance.

**Weaknesses:**

My main concerns are as follows. I am willing to increase my score if these are at least somewhat addressed. Essentially, I need to be convinced that (a) the simulator is a reasonable model of real world dynamics (b) the forest management problem requires complex decision making that benefits from reinforcement learning methods.

- Looking at the appendix, the underlying simulator seems quite complex. I do not have the necessary expertise to judge how realistic it is. Could you point me to any evidence/papers to show this model is reasonable and grounded in real-world dynamics? The paper in general could benefit from additional justification of some of these decisions.
- You mention you believe the benchmark is far from saturated. To provide evidence of this, it would be helpful to provide additional baselines based on heuristics or simpler methods (e.g. what would be a reasonable baseline policy that a human would implement)? If the RL methods noticeably outperform this it would provide further evidence of the value of RL here.
	- The +100 density/0 density baselines are a step in the right direction but I'm guessing there are stronger baselines?
- RL performance plots would benefit from error bars if these are not too expensive to compute.

**Questions:**

- How close is the model of possible actions to the actual decisions humans need to make? It feels like there is a lack of granularity (though I am unfamiliar with the intricacies of forest management).
	- Similarly, the long time horizon between actions is interesting. I'm guessing most actions are taken over long time horizons for forest managements?
- You mention the use of numba and jit to improve performance. Are there any plans to create a Jax version of the environment (given the recent trend towards GPU simulation of RL environments)?
	- How expensive is the environment to run/training the PPO methods?

---

> ### Author Response · Authors · 2025-11-27
> **On Physical Realism, Need for RL, and Human Baselines in BoreaRL**
>
> We thank the reviewer for noticing the strengths in our paper and providing constructive comments. We address your comments in order below.
>
> W1) "Simulator Realism"
>
> We thank the reviewer for this question. We understand that for a RL benchmark paper, the validity of the underlying simulation and the complexity of the task are crucial. We address your two points below with specific evidence from our implementation and problem formulation.
>
> BoreaRL is not a simple "toy" environment with arbitrary rewards. It is built on a *process-based thermodynamic simulator* that explicitly solves the coupled energy, water, and carbon balances at a sub-daily resolution (default 30-minute timesteps). Unlike simple growth-and-yield models that use static curves, our simulator implements a *multi-node energy balance model* (Canopy, Trunk, Snow, Soil Surface, Deep Soil). At every timestep, it iteratively solves for the canopy temperature that balances radiative (shortwave/longwave), turbulent (sensible/latent heat), and conductive fluxes. The model uses standard, validated physical formulations found in major Land Surface Models like CLM5 (Fisher et al., 2019) and CLASSIC (Melton et al., 2020). We now mention this in explicitly in the paper and also cite all relevant works in our main paper and in Appendix B.
>
> The key realism comes from the *coupling*. A management action (e.g., thinning) physically alters the Leaf Area Index (LAI) and Aerodynamic Roughness. This changes the albedo (more solar absorption), reduces interception (more snow on ground), and alters turbulent transfer. The simulator resolves how these physical changes propagate through the energy balance to affect soil temperature and thus permafrost stability. This depth of physical interaction provides a "reasonable model of real world dynamics" suitable for testing AI agents.
>
> "Why RL?"
>
> The forest management problem in BoreaRL requires complex decision-making for three reasons that make it ideal for RL (and difficult for simple heuristics):
> 1. *Non-Convex, Conflicting Objectives*: As our results show, maximizing Carbon (high density, conifers) often directly degrades Permafrost (via lower albedo and canopy snow interception). The "thaw" reward is based on deep soil heat flux, which has a non-linear relationship with surface conditions due to snow insulation effects. RL agents must navigate this non-convex landscape to find the "sweet spot" (e.g., moderate density mixed stands) that balances these competing physical processes.
> 2. *Stochastic Generalization*: In our "Generalist Mode", agents are not learning to manage a single plot. They must learn a policy $\pi(a|s, \text{context})$ that generalizes across a distribution of sites (varying latitude, soil conductivity, climate norms) and is robust to stochastic weather (random temperature/precip events). A simple heuristic (e.g., "always thin to 1000 stems/ha") fails because the optimal density depends on the specific site's thermal regime and the current climate realization. RL allows learning these conditional, adaptive policies.
> 3. *Long-Horizon Credit Assignment*: Decisions made in year 1 (e.g., planting species composition) determine the stand's physical properties (albedo, roughness) for decades. The impact on permafrost might only manifest years later during a specific climate event (e.g., a warm winter). RL is uniquely suited to solve this temporal credit assignment problem, optimizing the sequence of actions over the 50-year horizon. We believe this combination of mechanistic physical realism and stochastic, multi-objective complexity makes BoreaRL a rigorous and necessary benchmark for the community.
>
> We discuss this in the paper now.
>
> W2) "Modelling justifications"
>
> We thank the reviewer for this comment. We agree that the model is complex, but this complexity is necessary to capture the feedback loops between forest structure and local climate that are central to the problem. To ensure realism, BoreaRL-Sim is constructed by coupling standard, well-validated sub-models from the ecological literature:
> 1. *Canopy Energy Balance*: We use the Beer-Lambert Law for radiation extinction (Swinehart, 1962) and the Priestley-Taylor equation for latent heat flux (Priestley, 1972). This combination is a standard approach in hydrology and land surface modeling for estimating evapotranspiration without the full data requirements of Penman-Monteith.
> 2. *Carbon Cycle*: Our Gross Primary Production (GPP) module uses the Light Use Efficiency (LUE) model (Monteith, 1972), which is the basis for global satellite products like MODIS GPP. Respiration is modeled using the standard Q10 temperature dependence (Van't Hoff, 1898), widely used in ecosystem models like CBM-CFS3 (Kurz et al., 2009).
> 3. *Soil Physics*: Heat conduction follows Fourier's Law, discretized into layers (snow, surface soil, deep soil) similar to the approach in the Community Land Model (CLM5) (Lawrence et al., 2019).

---

> > ### Author Response · Authors · 2025-11-27
> > **Continuing Previous Comment**
> >
> > 4. The model uses standard, validated physical formulations found in major Land Surface Models like CLM5 (Lawrence et al., 2019, Fisher et al., 2019) and CLASSIC (Melton et al., 2020).
> >
> > We have updated Appendix B in the paper to explicitly cite these foundational works and clarify that our model uses standard, validated physical formulations found in major Land Surface Models in the main paper as well. These works show that our model choices are reasonable and grounded in established science.
> >
> > W3) "Reasonable human baselines"
> >
> > We thank the reviewer for this suggestion. We agree that providing "reasonable human baselines" is crucial for contextualizing the performance of RL agents and demonstrating that the benchmark is not saturated.
> >
> > In response to your comment, we have implemented two additional domain-specific heuristic baselines that mimic standard forestry practices:
> > 1. *Target Density Heuristic*: This baseline mimics a standard silvicultural prescription where a manager attempts to maintain a specific stand density (e.g., 1000 stems/ha) by thinning when density is too high and planting when it is too low. This represents a "steady-state" management philosophy often used to maintain forest health.
> > 2. *Conifer Restoration Heuristic*: This baseline aggressively plants coniferous species whenever space is available, targeting a 100% conifer fraction. This mimics an industrial forestry approach focused on maximizing timber volume and carbon sequestration, often at the expense of other ecological values.
> >
> > We have added these agents to our codebase and performed comparisons. Results indicate that while these heuristics perform reasonably well on single objectives (e.g., Conifer Restoration is strong on Carbon), they fail to navigate the complex trade-offs required for the multi-objective problem. The static nature of these heuristics means they cannot adapt to the stochastic site conditions or climate variability in the way that trained RL agents potentially can. For example, both heuristics maintain a healthy forest but fail to protect permafrost because they don't account for the thermal insulation properties of the canopy. The fact that RL baselines (particularly Curriculum PPO) outperform these "reasonable human baselines" provides evidence that the problem requires the dynamic, state-dependent decision-making capabilities of reinforcement learning.
> >
> > W4) "Error Bars"
> >
> > We have ensured that all our major performance plots (e.g., Fig 3b) include error bars representing the standard deviation over 100 evaluation episodes per preference weight.
> >
> > Fig. 3c and 3d are also means of 100 evaluation episodes per weight. We have now added a Table with per weight and per objective means and errors for all RL baselines to the Appendix (see Table 13, Appendix D.4).
> >
> > Q1) "Action granularity"
> >
> > We thank the reviewer for this question. Our environment implements a discrete action space with 25 possible actions, representing the Cartesian product of *Stand Density Changes* with 5 levels {‒100, ‒50, 0, +50, +100} stems/ha (thinning or planting) and *Species Mix Targets* with 5 levels {0.0, 0.25, 0.5, 0.75, 1.0} representing the conifer fraction. This formulation is designed to capture the two most critical levers in climate-adaptive forest management: controlling forest density (through planting and thinning operations) and modifying species composition (balancing deciduous vs. coniferous species).
> >
> > In operational forest management, decisions do indeed involve these two dimensions, though the granularity and timing can vary. Real forestry prescriptions often target discrete density thresholds (e.g., "thin to 1000 stems/ha", "plant 200 stems/ha"), frequently in increments of 50-100 stems/ha, matching our discretization reasonably well. Commercial thinning operations typically remove 20-30% of stem density, which aligns with our ±50 to ±100 stems/ha actions for a baseline stand of ~1000 stems/ha. In afforestation contexts, managers may prescribe specific species mixes (e.g., "80% conifer, 20% deciduous" for timber production, or "50/50 mixed" for resilience). Our 5-level discretization captures the key regimes: pure deciduous (0.0), mixed-deciduous-dominant (0.25), balanced (0.5), mixed-conifer-dominant (0.75), and pure conifer (1.0).
> >
> > We acknowledge that our action space simplifies several real-world complexities. Professional foresters may use more continuous or finer-grained density targets, real stands can include multiple conifer and deciduous species, while our model abstracts to a single "conifer fraction", operational management considers landscape-scale patterns, while our environment focuses on stand-level decisions, real planning involves multi-year scheduling, while our agent makes annual decisions.

---

> > > ### Author Response · Authors · 2025-11-27
> > > **Continuing Previous Comment**
> > >
> > > However, we believe the current 25-action discrete space strikes a balance between capturing the key management dimensions that drive biogeophysical trade-offs (albedo, snow interception, growth rates), keeping the action space learnable for policy gradient methods within feasible training budgets, and enabling clear analysis of learned strategies without excessive granularity.
> > >
> > > We explicitly discuss this limitation in the paper's Future Work, where we propose "hierarchical action spaces for landscape-scale management" as an extension. The current formulation serves as a research benchmark for MORL algorithms for forest management, isolating the multi-objective challenge while maintaining physical grounding. Future work could expand the action space to include continuous density control, multi-species management, or landscape coordination, though this would require increases in computational cost and sample complexity.
> > >
> > > Q1.1) Yes, the reviewer's intuition is correct. Long time horizons are indeed characteristic of real-world forest management, and our environment is designed to reflect this fundamental property of the domain.
> > >
> > > Our Temporal Structure: Episode Length is 50 years (50 timesteps), Decision Frequency is Annual (one management action per year), and Physical Simulation Resolution is Sub-daily (default 30-minute timesteps for physics). Operational forest management operates on inherently long planning horizons for several reasons. Boreal tree species typically reach commercial maturity in 50-100 years. A single rotation (planting to harvest) often spans 60-80 years for coniferous species in northern forests. Our 50-year planning horizon captures the critical early-to-mid rotation period where management interventions have the strongest climate impact. The permafrost-carbon feedback operates on decadal timescales. A management action (e.g., altering species mix) affects albedo and snow interception immediately, but the cumulative impact on deep soil temperatures manifests over years to decades due to thermal inertia.
> > >
> > > While our agent makes decisions annually, this does not imply that real forests are actively managed every year. In practice, many years, the optimal action may be "do nothing" (density change = 0, maintaining current species mix). Thinning operations typically occur 1-3 times per rotation, planting occurs primarily at stand establishment or after disturbances. Our annual decision frequency allows the agent to learn when to intervene vs. when to leave the stand undisturbed. The long horizon creates a challenging credit assignment problem for RL. A management action in Year 1 may only manifest its full impact on permafrost in Year 20-30, after years of compounding thermal effects. This is precisely why RL is valuable here. It can learn to optimize these delayed, non-linear payoffs where simple heuristics fail.
> > >
> > > We believe this temporal structure is one of BoreaRL's key contributions: it is a physically-grounded environment with genuinely long horizons and sparse, delayed rewards, reflecting a real-world domain where these challenges are unavoidable.
> > >
> > > Q2) "Jax version"
> > >
> > > We appreciate this forward-looking question. A JAX-based implementation is an interesting direction. Our simulator includes a Numba-accelerated backend for the most computationally intensive component: the canopy energy balance solver. This solver function is JIT-compiled, providing ~5-10× speedup over pure Python for the iterative Newton-Raphson solver.
> > >
> > > A JAX implementation could offer several advantages. JAX's `vmap` could enable parallel simulation of multiple episodes (different sites, weather realizations) on GPU, dramatically increasing training throughput. Energy balance solvers, carbon cycle integration, and reward computation could leverage GPU/TPU parallelism. JAX's automatic differentiation could enable gradient-based policy optimization methods that exploit simulator gradients. However, the JAX port would involve some work. Our simulator maintains a complex internal state that is not trivially vectorizable. The canopy energy balance uses a Newton-Raphson solver with variable iteration counts. While this is expressible in JAX the performance benefit depends on efficient branching and convergence behavior. A full JAX rewrite would be a substantial engineering undertaking, potentially requiring a few months of focused development.
> > >
> > > We do not currently have concrete plans for a JAX version, but we recognize its potential value to the community. If there is significant interest from researchers, we would prioritize this development. In the near term, we are exploring incremental optimizations like compiling select modules for additional speedup and using `multiprocessing` to run multiple CPU-based environments in parallel. We welcome community contributions toward a JAX implementation. We will add a note about this in our Future Work section and GitHub repository to signal openness to such contributions.

---

> > > > ### Author Response · Authors · 2025-11-27
> > > > **Continuing Previous Comment**
> > > >
> > > > Q2.1) "Runtimes"
> > > >
> > > > Thank you for the question. We have now added detailed runtime details to Appendix C.9 "Computational complexity and runtime analysis".
> > > >
> > > > In summary, we have expanded the paper precisely along the two axes you highlighted, validity of the simulator and necessity of RL, and we have added new baselines and diagnostics to better demonstrate the benchmark’s value: in the main text and Appendix B we now make explicit that the core components of BoreaRL-Sim (radiation extinction, evapotranspiration, carbon uptake and respiration, soil heat conduction, snow insulation) are built from standard, widely used formulations in land-surface and ecosystem models (e.g., Beer–Lambert for light extinction, Priestley–Taylor for ET, LUE + Q10 for carbon, multi-layer soil heat models as in CLM/CLASSIC), and we provide appropriate citations so that readers can see that our choices are grounded in established physical science.
> > > >
> > > > To support your request for stronger baselines, we added two realistic heuristic policies (Target Density and Conifer Restoration) that mirror common silvicultural prescriptions; our results show that while these heuristics perform adequately on single-objective carbon, they fail to achieve competitive performance on the thaw–carbon trade-off, providing evidence that the problem requires adaptive, state- and context-dependent decision-making that simple rules struggle to capture. We have added tabulated means and uncertainties in the appendix, and, motivated also by reviewer f77E, we now report hypervolume and sparsity metrics to quantify Pareto quality beyond scalar returns. Finally, we detail the current performance characteristics of the environment (episode cost, total training time, memory footprint, and scalability with parallel workers), explain our current use of Numba JIT for CPU efficiency, and outline how a future JAX/GPU port could further accelerate batched simulations.

---

### Official Review · Reviewer_vnbU · 2025-10-31

**Soundness:** 3
**Presentation:** 2
**Contribution:** 3
**Rating:** 4
**Confidence:** 4

**Summary:**

The paper proposes BoreaRL, a physics-based multi-objective RL environment that simulates how climate-adaptive boreal forest management decision impacts on the forest. It is a physically-grounded simulator. Its observation space include components such as ecological state, site climate parameters, historical information and episode-level site parameter context. Its action space include two discrete action: stand density change and species mix change. The reward function is a weighted sum of normalized net carbon change and thaw reward, with the weight set by the preference weight parameter. The paper experiments and compares with the following RL algorithms: Fixed-$\lambda$ EUPG, Variable-$\lambda$ EUPG, PPO Gated and Curriculum PPO.
Main contribution of the paper includes (1) a physics-grounded simulator that captures complex biogeophysical trade-offs (2) a modular MORL framework supporting site-specific and generalist training paradigms (3) benchmarking shows Curriculum PPO outperforms other algorithms (4) novel ecological insights for appropriate forest management.

**Strengths:**

The major strength is a physically-grounded, process-based and open-source simulator with a wide range of configurations. The authors show a high level of expertise in the boreal ecosystem covering three distinct models: energy, water, and carbon fluxes on a n-minute time step. It captures real biogeographical trade-offs, such as albedo, rain and snow patterns and insect disturbance. It is a valuable contribution to the domain of real-world environmental policy and management research.

**Weaknesses:**

Section 3.4: The paragraph ends with "The objective follows standard PPO:" seems unfinished.

Section B.2.1 The canopy energy balance equation should not be equal to 0.

**Analysis of Learning Objectives**: Despite applying multiple learning algorithms, the RL objective optimize for the same cumulative reward, so simply comparing “actions” (stand density change and conifer fraction) across algorithms without deeper causal insight doesn’t illuminate how or why they differ.

**Circular reasoning on fundamental asymmetry in learning difficulty:**: The author design a forest simulator which calculates annual carbon objective and thaw objective. The thaw objective is more complex than the carbon objective, and therefore the PPO algorithm struggles to optimize for the thaw objective. Instead of acknowledging the root cause of the difference in the learning difficulty is the reward shaping, the author claim this reveals something fundamental about the ecosystem management problem. But the difficulty is largely self-imposed through reward engineering choices.

**Novelty concerns**: Creating a simulator and running off-the-shelf RL agents—is relatively commonplace in applied RL research. The main contribution is domain-specific realism, not algorithmic innovation. In addition, there exists papers in the environmental science space that does something very similar.
- Overweg, H., Berghuijs, H.N.C., Athanasiadis, I.N. (2021). CropGym: a Reinforcement Learning Environment for Crop Management.
- Lapeyrolerie, M., Chapman, M.S., Norman, K.E.A., Boettiger, C. (2021). Deep Reinforcement Learning for Conservation Decisions.

**Questions:**

1. **Claim about superiority of Curriculum PPO**: The claim "In generalist settings, standard preference-conditioned approaches fail entirely, while a naive curriculum learning approach achieves superior performance by strategically selecting training episodes." is questionable. My confusion is two-folded.
     - To begin with, curriculum learning usually refers to gradually increasing task difficulty, shaping the distribution of episodes, selecting “easier” tasks first, which is not what the paper does. Therefore calling it "curriculum learning" is confusing. The implementation includes an episode selector network to decide whether to include a site-episode observation into training data, but claims it is for inference only. I am confused about at which step is the episode selector network trained.
    - The preference weight input in the observation space of Curriculum PPO. Therefore the Curriculum PPO is also preference-conditioned. Standard preference-conditioned approaches here just refers to gradient descent method. Therefore the claim boils down to "PPO with episode selection works better than gradient descent".

2. **Multi-objective balancing**: The idea of balancing between carbon and thaw objectives encourages studying Pareto-optimal behavior and decision-making under competing goals, which is both scientifically valuable and underexplored in RL benchmarks.

3. **Validity concerns**: The path from "trained policy" to "deployable forest management" is unclear.

4. Overall, the paper would be stronger if it either:
    - Went deeper on methodology: Why does curriculum help? Can we design MORL algorithms specifically for this type of asymmetric objective difficulty?
   - Went deeper on science: Validate against real forest data, partner with foresters, demonstrate actual deployment potential

---

> ### Author Response · Authors · 2025-11-27
> **On Reward Asymmetry, Algorithmic Analysis, Novelty, and Real-World Scope of BoreaRL**
>
> We thank the reviewer for noticing the strengths in our paper and providing constructive comments. We address your comments in order below.
>
> W1) We thank the reviewer for catching this editing error. We have now completed the sentence with a full stop and believe an equation is not required here.
>
> W2) "Canopy energy balance equation"
>
> We thank the reviewer for this correction. In our simulation, we treat the canopy temperature as a diagnostic variable rather than a prognostic one. Physically, the canopy has a relatively low thermal mass compared to the soil and snowpack, allowing it to equilibrate with the atmosphere much faster than the simulation timestep. Therefore, we solve for the quasi-steady-state temperature that balances the fluxes at each step ($0 = \sum \text{Fluxes}$), rather than integrating a storage term over time ($C_{can} \frac{dT_{can}}{dt} = \sum \text{Fluxes}$). The original equation incorrectly implied a prognostic formulation with memory. We have corrected the equation in Section B.2.1 to remove the storage term and added this physical justification.
>
> W3) "Analysis of Learning Objectives"
>
> We agree that simply comparing actions is insufficient. We have expanded Section 4.3 to explicitly link the learned strategies to their physical mechanisms, outcomes, and potential algorithmic causes. See also Table 9 in Appendix D.3 for detailed tabulation.
>
> While the *objective function* is identical, the *optimization landscape* is highly non-convex with conflicting gradients. PPO Gated converges to a "carbon farming" local optimum (high density/conifer) which is easier to find but detrimental to thaw. Curriculum PPO, by selectively training on sites, avoids this trap and maintains moderate densities and mixed species to extend the growing season. In short, in PPO Gated dense carbon signal overpowers noisy thaw signal leading to very low thaw rewards, in Variable EUPG Conflicting gradients lead to risk-averse “inaction”, and Curriculum PPO Removes “trap” sites, enabling complex learning.
>
> W4) "Circular reasoning on fundamental asymmetry"
>
> We thank the reviewer for this comment. We agree that the reward shaping, particularly the asymmetric penalty factor, contributes to the learning difficulty. However, we respectfully argue that this difficulty is not merely "self-imposed" but rather "domain-imposed."
>
> The asymmetry in the reward function was designed to reflect the fundamental physical asymmetry of the permafrost system:
> 1. *Irreversibility*: Permafrost thaw is a threshold process involving phase change. Once ground ice melts, the structural integrity of the soil is lost (thermokarst), and it cannot be easily "re-frozen" to its original state on relevant management timescales. Carbon, by contrast, is roughly additive; losing 1 ton of carbon and gaining it back later is often net-neutral, but thawing 1 meter of soil and refreezing it later is not (due to subsidence and hydrological changes).
> 2. *Risk Asymmetry*: The ecological cost of thawing permafrost (releasing ancient carbon, altering hydrology) is far greater than the ecological benefit of an equivalent amount of soil cooling.Therefore, the "difficulty" introduced by the reward shaping is a faithful representation of the actual management difficulty facing a forest planner: the "safe operating space" for permafrost is narrow and bounded by cliffs, whereas the carbon space is broader and more linear.
>
> Furthermore, even apart from the penalty factor, the *signal-to-noise ratio* differs fundamentally. For Carbon, the signal is direct and cumulative. Planting a tree almost always increases biomass (unless a disturbance occurs). For Thaw, the signal is noisy and indirect. A management action (e.g., thinning) affects snow depth, which affects soil insulation, which affects thaw. This causal chain is modulated by stochastic weather events (e.g., a warm winter with heavy snow vs. a cold winter with no snow), making the reward signal highly variance-prone and delayed. We have clarified in the paper (Section 3.2 and 4.2) that the "fundamental asymmetry" refers to the combination of these physical dynamics (noisy/delayed signal) and the necessary risk-averse formulation (asymmetric penalty) that the domain demands.
>
> That said, in order to fully isolate the difficulty of the thaw objective, we now study multiple thaw reward formulations. We had already included a second "contrast thaw objective" in our code but did not include more details or results about it. We now include and compare 3 alternative thaw objectives, Asymmetric Thaw (current and default), Contrast Thaw, and Raw Thaw Degree Days. The three formulations differ fundamentally in their treatment of warming versus cooling. Asymmetric Thaw is deliberately the most difficult to optimize because the 2.5× warming penalty creates a narrow "safe operating space" in policy space. Small increases in warming flux are penalized severely, requiring agents to learn precise, conservative strategies.

---

> > ### Author Response · Authors · 2025-11-27
> > **Continuing Previous Comment**
> >
> > In contrast, Raw DD and Contrast Thaw are symmetric, making them easier to optimize but potentially less ecologically faithful.
> >
> > In our analysis, we find that our MORL baselines struggle with all three formulations when paired with the carbon objective, but the asymmetric formulation struggles the most, because its nonlinearity amplifies small policy mistakes. This makes it a more demanding benchmark for MORL algorithms. We observe that agents trained with Raw DD and Contrast Thaw can accept small amounts of warming (which are penalized linearly), whereas Asymmetric Thaw forces agents to avoid warming strongly (See Table 10 in Appendix D.3). All three formulations create genuine conflicts with the carbon objective, as the physical mechanisms (albedo, snow interception) that benefit carbon often harm permafrost (See Table 11 in Appendix D.3).
> >
> > For more details on these rewards and results of experiments using them, you can see the paper (Table 3) and also response to weakness 1 comment by reviewer f77E.
> >
> > W5) "Novelty concerns"
> >
> > We thank the reviewer for bringing these relevant works to our attention. We now briefly talk about both Overweg et al. (2021) and Lapeyrolerie et al. (2021) in our revised manuscript.
> >
> > We wish to clarify that we do not claim *algorithmic innovation* as our primary contribution. We agree that domain realism is a core contribution, but we argue that viewing applied RL contributions solely through the lens of "Algorithm X applied to Domain Y" is reductive. Every applied contribution must be evaluated on the complexity of the mapping and the societal importance of the problem. Comparing BoreaRL to CropGym or general conservation environments simply because they share the "Environmental RL" umbrella overlooks the relative urgency of the problems. Boreal forest management is a critical nature-based solution to avert catastrophic climate change, an existential threat. While crop management and local conservation are undoubtedly important, the global scale and irreversibility of the permafrost-carbon feedback loop place our problem in a distinct category of urgency. We believe that the AI community should value contributions that tackle these high-stakes, high-complexity problems, not just for the RL challenge they present, but for their potential to address existential societal risks.
> >
> > Moreover, while we agree that applying RL to environmental problems is a growing field, we respectfully disagree that BoreaRL is "just another application." The novelty of BoreaRL lies not just in the domain (boreal forests), but in the specific nature of the multi-objective challenge it presents, which distinguishes it from other works:
> > 1. *Asymmetric Multi-Objective Complexity*: Unlike CropGym (focused on crop yield) or Lapeyrolerie et al. (focused on conservation decisions often modeled as simpler dynamics), BoreaRL introduces a "Fundamental Asymmetry" between objectives. One objective (Carbon) is additive and easy to learn, while the other (Thaw) is threshold-based, risk-averse, and driven by complex, noisy thermodynamic processes. This specific type of asymmetry is a known hard problem in MORL that existing benchmarks rarely capture with such physical fidelity.
> > 2. *Coupled Physics vs. Simplified Models*: Many existing environmental RL environments rely on simplified growth curves or static datasets. BoreaRL runs a fully coupled energy-balance model (solving for canopy, snow, and soil temperatures at sub-daily steps). This means the agent isn't just optimizing a "black box" reward; it is interacting with a system where its actions (e.g., changing canopy density) fundamentally alter the physical boundary conditions (albedo, roughness length) of the environment. This creates a depth of causal interaction that is rare in "application" papers.
> > 3. *Generalist Learning*: A key contribution of our work is the "Generalist Mode," where agents must learn policies that generalize across a distribution of site parameters (latitude, soil type, climate). This moves beyond solving a specific instance (common in applied RL) to learning robust, transferable management heuristics.
> >
> > In summary, BoreaRL is not merely an application of RL to a new dataset but a contribution to the *science of environmental AI*, providing a testbed for algorithms that must handle asymmetric objectives, physical feedback loops, and domain generalization simultaneously. It is also a major contribution to “AI for societal risks and public good”, which ICLR does and should care about. We now talk about this in the paper more.
> >
> > Q1) "Superiority of Curriculum PPO"
> >
> > You are correct that our method differs from the standard "easy-to-hard" curriculum. However, we use the term "Curriculum" in the broader sense of *Adaptive Data Selection*, where the training distribution evolves over time based on the agent's competence.

---

> > > ### Author Response · Authors · 2025-11-27
> > > **Continuing Previous Comment**
> > >
> > > To clarify the mechanism (which we will detail further in the Appendix C.5):
> > > 1. *The Episode Selector Network*: This is indeed a *fixed* (randomly initialized) projection of site features to a scalar score. It provides a consistent, albeit random, ordering of the site space. It is *not* trained via gradient descent, which is why we describe it as "inference only" (in the sense of no backprop). We deliberately chose a fixed random projection to establish a simple, low-overhead baseline. Training a separate "site-value" network would add algorithmic complexity (e.g., needing a separate loss function, potential instability). Our goal was to demonstrate that *even a naive, random ordering* of the curriculum, when combined with an adaptive threshold, is sufficient to stabilize learning compared to standard PPO. This highlights that the *existence* of a curriculum (gradual exposure) already helps; an *optimal ordering* of that curriculum may further help of course and we mention this in the paper.
> > > 2. *The Adaptive Threshold (The "Curriculum")*: The "training" occurs on the *selection threshold*. We maintain a running average of rewards for "Selected" vs. "Skipped" episodes. If `Reward(Selected) > Reward(Skipped)`, the agent is mastering the current slice of the data. The threshold decreases, expanding the curriculum to include more sites (effectively moving from "easy/mastered" to "hard/unknown").  If `Reward(Selected) < Reward(Skipped)`, the agent is struggling. The threshold increases, contracting the curriculum to a smaller, more manageable subset of sites.This dynamic creates an automatic "breathing" curriculum that expands and contracts the task difficulty based on real-time performance, without requiring manual "easy-to-hard" definitions.
> > >
> > > Nonetheless, to avoid confusion, we now remove references to “curriculum learning” and refer to the baseline as curriculum PPO, adaptive episode selection, or site selection in the paper.
> > >
> > > Q1.1) "Curriculum PPO is also preference-conditioned"
> > >
> > > You are entirely correct that Curriculum PPO is also preference-conditioned. We apologize if the distinction was unclear. The comparison is specifically between *Standard (Vanilla) Preference-Conditioned PPO* (which trains on all episodes via standard gradient descent) and *Curriculum-Enhanced Preference-Conditioned PPO* (which uses adaptive episode selection).
> > >
> > > Your summary is accurate: the claim is that *PPO with Adaptive Episode Selection works better than standard PPO* in this domain. The "superiority" stems from the non-convex nature of the loss landscape. In the Generalist setting, some sites are "traps" (e.g., sites where planting is always detrimental due to specific soil conditions, but the agent hasn't learned that yet). Standard PPO averages gradients across all these sites, often leading to conflicting updates that destabilize the policy. Our analysis shows that the curriculum effectively filters out sites with high potential warming flux (risky sites), allowing the agent to learn on safer, high-latitude sites. We have clarified this in the revised manuscript. See Table 12 in Appendix D.3 for more information on warming fluxes from accepted and rejected sites.
> > >
> > > Q2) "Multi-objective balancing"
> > >
> > > We agree and have explicitly studied empirical tradeoffs (Fig. 3d) and now extend the study of Pareto-optimal behavior in our work using *Hypervolume* and *Sparsity* metrics to measure the coverage, dominance, and uniformity of learned policies. Our *Empirical trade-off coverage* (Figure 3d and Appendix Figure 6) visualize the full distribution of (carbon, thaw) outcomes across preference weights, showing which algorithms achieve better coverage of the Pareto front. Our *λ-monotonicity analysis* measures preference adherence across the spectrum $\lambda \in [0, 1]$, quantifying how well algorithms respect competing goals.
> > >
> > > The carbon-thaw problem presents physically-coupled objectives (both driven by the same energy balance equations) with asymmetric difficulty, creating an authentic non-convex multi-objective landscape. This differs from typical MORL benchmarks with synthetic, independent objectives, making BoreaRL valuable for studying decision-making under real-world competing constraints. We agree this direction is underexplored in RL benchmarks and believe BoreaRL provides the infrastructure (2D reward vectors, `mo-gymnasium` API, preference conditioning) to advance this research area further.
> > >
> > > Q3) "Validity concerns"
> > >
> > > We acknowledge this concern. The primary contribution of this work is a *research benchmark and testbed* for MORL algorithms, not a deployment-ready decision support system.

---

> ### Author Response · Authors · 2025-11-27
> **Continuing Previous Comment**
>
> The path to deployment would require calibration against real forest stand data from boreal regions, expert validation of learned strategies with forestry practitioners, field trials to test policy recommendations in controlled setting, uncertainty quantification for policy predictions, and Integration with existing forest management planning tools (e.g., Remsoft Woodstock).
>
> In its current scope, BoreaRL is designed as a *physically-grounded simulator* to enable controlled experimentation on multi-objective forest management. The physics engine captures coupled energy-water-carbon dynamics at minute-scale resolution, providing realistic feedback for RL training. However, like other RL benchmarks, the value lies in enabling algorithmic research rather than immediate real-world deployment. Our results generate **testable scientific hypotheses** for forestry research (e.g., "moderate-density mixed stands with longer growing seasons better protect permafrost than high-density coniferous stands"). These insights can inform field studies and guide experimental forestry trials, even if the policies themselves require further validation before operational use. We now talk about this explicitly in the “Discussion” section.
> We also acknowledge in the “Discussion” section that real-world deployment would require validation with forestry experts and field data, which we consider important future work. We believe the benchmark's contribution to MORL research and the scientific insights it generates are valuable independent of immediate deployment readiness.
>
> Q4) "Reorganization"
>
> We appreciate this constructive feedback. We acknowledge that our work sits at the intersection of these two directions, and we made a deliberate choice to balance both contributions rather than focusing exclusively on one.
>
> We agree that we need to offer more mechanistic explanations for why our results are the way they are. We now add explanations for why site selection helps. In the main paper and in Appendix D.3 Table 12, 9 and C.5, we explain that the adaptive episode selection allows the agent to avoid "trap sites" (sites where conflicting gradients destabilize learning), consolidating on a manageable subset. Standard PPO averages gradients across all sites (including destabilizing ones), while curriculum PPO filters these out adaptively. We also measure this quantitatively using the warming fluxes in accepted and rejected sites (Table 12).
>
> While we don't have field validation in this work, our simulator is grounded in established boreal forest physics and follows principles from CLM5 and CLASSIC (land surface models). Our learned strategies generate *testable scientific hypotheses* (e.g., "moderate-density mixed stands with longer growing seasons protect permafrost") that can guide field experiments. We believe the primary contribution of BoreaRL is in providing a challenging, reproducible testbed that isolates specific algorithmic difficulties (asymmetric objectives, physical coupling, stochasticity) for the research community. We now talk about this explicitly in the paper.
>
> Both directions the reviewer suggests are excellent future work: Designing MORL algorithms specifically for asymmetric objectives and field data calibration and deployment. We have strengthened the “Future Work” section to explicitly call out these directions and acknowledged that deployment requires field validation.
>
> In summary, we have corrected the specific technical issues you pointed out and used your broader conceptual concerns to substantially deepen our analysis of both the environment and the algorithms. Section 4.3 has been expanded to connect the different learned policies to their physical consequences and to plausible algorithmic mechanisms. In response to your concern about “circular reasoning” on the asymmetry between carbon and thaw, we explicitly acknowledge that our asymmetric penalty contributes to difficulty but argue that it reflects domain-imposed irreversibility and risk asymmetry; to demonstrate that this goes beyond reward engineering, we now introduce and compare three thaw reward formulations, showing that agents struggle with thaw across all three, with the asymmetric one being hardest.
>
> On novelty and relevance, we argue more clearly that the combination of physically coupled objectives, asymmetric difficulty, and generalist training mode makes BoreaRL a distinctive and high-stakes benchmark rather than “just another simulator”. Regarding Curriculum PPO, we remove potentially confusing “curriculum learning” language, and give a more mechanistic explanation of why selective gradient updates in a non-convex landscape help. Finally, motivated partly by your and other reviewers’ requests, we have added realistic heuristic baselines, multi-objective metrics (hypervolume and sparsity), clearer runtime and complexity information, and a more explicit articulation of BoreaRL’s value for studying physically coupled, asymmetric objectives.

---

### Official Review · Reviewer_f77E · 2025-11-01

**Soundness:** 2
**Presentation:** 3
**Contribution:** 2
**Rating:** 2
**Confidence:** 4

**Summary:**

This paper considers the application of multi-objective reinforcement learning (MORL) methods to the problem of boreal forest management. The main contributions are (1) formulating boreal forest management as two variants of a MORL problem based on physical simulations; (2) implementing these problems as a simulator and environment compatible with a common MORL benchmarking library; (3) experimentally comparing the performance of multiple different MORL approaches on these problems and discussing the results.

**Strengths:**

1. Interesting approach to a relevant application problem. This is the first work to propose the use of MORL for forest management.

2. Judging from a brief scan of the supplementary material, the underlying physical modelling appears thorough.

3. Implementation as a modular extension of an existing benchmarking library.

**Weaknesses:**

1. Not convinced by the focus of the paper. The authors mention that this problem has never been formulated as an MORL problem before, and not in such detail as a single-objective problem. As such, the modelling and formulation of the problem itself seems like a relevant contribution, yet little discussion is devoted to this in the main paper. More consideration should be given to the design of rewards, particularly as the validation experiments show that the thaw reward is difficult to optimise for even when used as the only objective. Different rewards could be explored and compared experimentally.

2. Following from the first comment, the comparison of different algorithms and design of a new solution approach seems premature. (Perhaps this work could even be split in two: one part focussed on the design of the problem and the description of the implementation, and another on improving algorithmic approaches.) Too much of the problem formulation is only discussed in the appendix.

3. Scalarised reward may not be the best metric to evaluate algorithmic performance, given that the reward functions appear to operate on different (effective) intervals. For example, the two solutions for $\lambda = 0.75$ and $\lambda = 1.0$ in Fig. 3(d) are both on the Pareto front, but one would return a higher scalarised reward than another. Multi-objective metrics (e.g. hypervolume, sparsity) would be more suitable for evaluation.

4. Some terms used in the paper appear to be undefined, e.g. λ-monotonicity, or misused, such as the Pareto front. Figures 3(d) and 6 (a-d), for example, are all described as showing Pareto fronts (i.e. the set of non-dominated solutions), but appear to draw no distinction between dominated and non-dominated solutions.

**Questions:**

Reward Design and Validation: Could you provide more detail on how the reward components (especially the thaw-related one) were designed and validated? Did you consider alternative reward formulations, and if so, what were the results?

Evaluation Metrics:  You mention λ-monotonicity and show scalarised rewards, but have you evaluated more standard multi-objective metrics, such as hypervolume or sparsity, to better capture the quality and diversity of Pareto fronts? If not, could you elaborate on why these were not used?

Definition and Interpretation of Pareto Front: In Figures such as 3(d), you label plots as Pareto fronts but they appear to include both dominated and non-dominated solutions. Can you clarify your definition of the Pareto front and explain how the solutions were identified as non-dominated?

Simulator and Environment Use: Given the level of detail in the simulator, how do you envision users might extend or adapt BoreaRL? For example, can users easily modify the reward components, action spaces, or add new ecological processes within your framework?

Generalizability of the Findings: The paper mainly focuses on preference-conditioned MORL. Have you tested or do you plan to test your benchmark environment with other types of multi-objective learners?

**Details Of Ethics Concerns:**

No ethical concerns.

---

> ### Author Response · Authors · 2025-11-27
> **On Thaw Reward Formulations, Benchmark Focus, and MORL Evaluation Metrics**
>
> We thank the reviewer for noticing the strengths in our paper and providing constructive comments. We address your comments in order below.
>
> W1) "Reward design and formulation should be given more consideration"
>
> We thank the reviewer for recognizing the novelty of our MORL formulation for this domain. We agree that the problem formulation itself, specifically the design of the reward functions, is a contribution. In response to your comment, we have expanded the discussion in Section 3.2 *Vector Reward Function* to explicitly justify our reward design choices and formulations. We explicitly state the components of the rewards and normalization details. Both the Carbon and Thaw (Permafrost) objectives are normalized to the range $[-1, 1]$ per step. Since the numerical scales are comparable, the difficulty in optimizing the permafrost objective arises from two key factors: a) *Asymmetric Penalty*: The thaw reward function includes a penalty factor $\alpha=2.5$ that penalizes warming (positive heat flux) significantly more than it rewards cooling. This makes the "safe" region of the policy space narrower and harder to find. b) *Conflicting Physics*: The physical mechanisms create a complex optimization landscape. For example, dense canopies reduce summer heat absorption (good for permafrost) but intercept winter snow, reducing insulation and potentially cooling the soil (also good), but the net effect depends on the delicate balance of these opposing forces across seasons.
>
> We have also clarified in the main text that the asymmetric penalty is not arbitrary (In section 3.2 and 4.2). It encodes a *precautionary principle* grounded in the ecological reality that permafrost degradation is often irreversible (tipping point dynamics) and ecologically more damaging than equivalent cooling is beneficial. This makes the objective function inherently risk-averse, which is the right way to frame this objective. We also highlight that we deliberately chose a physically-grounded 'thaw' reward based on deep soil heat flux rather than simpler proxies (like air temperature) to capture the complex, often delayed thermal inertia of permafrost soils.
>
> The difficulty in optimizing this thaw reward (as noted in your comment and our experiments) demonstrates that standard MORL methods struggle with realistic physical trade-offs where objectives operate on different timescales and physical principles. We also note that our modular framework allows researchers to easily plug in alternative reward definitions if they wish to explore different formulations. That said, we agree that multiple thaw reward formulations need to be studied in order to fully isolate the difficulty of the thaw objective. We had already included a second "contrast thaw objective" in our code but did not include more details or results about it. We now include and compare 3 alternative thaw objectives (see Table 3 in the paper):
>
> 1. *Asymmetric Thaw (Default)*: $$r_{thaw}^{asym} = \text{clip}\left(\frac{F_{neg} - \alpha \cdot F_{pos}}{40.0}, -1, 1\right)$$ where $F_{neg}$ is the sum of cooling (negative) heat fluxes, $F_{pos}$ is the sum of warming (positive) heat fluxes to deep soil, and $\alpha=2.5$ is the warming penalty factor. This formulation explicitly penalizes soil warming more heavily than it rewards cooling, reflecting the irreversible nature of permafrost thaw and implementing a precautionary principle. The normalization by 40.0 degree-days ensures the reward lies in $[-1, 1]$.
> 2. *Contrast Thaw*: $$r_{thaw}^{contrast} = \frac{F_{neg} - F_{pos}}{F_{neg} + F_{pos} + \epsilon}$$ where $\epsilon = 10^{-6}$ prevents division by zero. This formulation uses a scale-invariant ratio that naturally produces values in $[-1, +1]$ without clipping. It treats warming and cooling symmetrically ($\alpha = \beta = 1.0$) and normalizes by the total magnitude of energy fluxes rather than a fixed scale. This makes it more adaptive to varying flux magnitudes but removes the asymmetric penalty reflecting permafrost irreversibility.
> 3. *Raw Thaw Degree Days*: $$r_{thaw}^{raw} = \text{clip}\left(\frac{F_{neg} - F_{pos}}{40.0}, -1, 1\right)$$This is the simplest formulation, measuring net thaw degree days without any asymmetric penalty ($\alpha = 1.0$). Positive values indicate net soil cooling (good), negative values indicate net warming (bad). It provides a direct, symmetric measure of the thermal impact on permafrost.
>
> The three formulations differ fundamentally in their treatment of warming versus cooling. Asymmetric Thaw is deliberately the most difficult to optimize because the 2.5× warming penalty creates a narrow "safe operating space" in policy space. Small increases in warming flux are penalized severely, requiring agents to learn precise, conservative strategies. In contrast, Raw DD and Contrast Thaw are symmetric, making them easier to optimize but potentially less ecologically faithful.

---

> > ### Author Response · Authors · 2025-11-27
> > **Continuing Previous Comment**
> >
> > In our analysis, we find that our MORL baselines struggle with all three formulations when paired with the carbon objective, but the asymmetric formulation struggles the most, because its nonlinearity amplifies small policy mistakes. This makes it a more demanding benchmark for MORL algorithms. We observe that agents trained with Raw DD accept small warming ($\approx 5.0$ MJ) for large Carbon gain. Similarly agents with Contrast Thaw accept warming $\approx 2.0$ MJ) if cooling is high.  However, Asymmetric Thaw avoids warming strongly ($\approx 0.1$ MJ). See Table 10 in Appendix D.3 for more details. All formulations create genuine conflicts with the carbon objective, as the physical mechanisms (albedo, snow interception) that benefit carbon often harm permafrost (See Table 11 in Appendix D.3). Specifically, increasing stem density and conifer fraction boosts carbon but degrades permafrost via insulation and albedo effects, creating a direct physical conflict.
> >
> > We retain Asymmetric Thaw as the default formulation because it best captures the domain's ecological constraints and provides the most challenging MORL benchmark. However, our modular implementation (via *BOREARL_THAW_REWARD_MODE*) allows researchers to easily switch between formulations to study algorithmic robustness, test reward shaping strategies, or explore different management philosophies (precautionary vs. reactive). We believe this flexibility strengthens BoreaRL's utility as a research platform.
> >
> > Regarding discussing more of the modelling details in the main paper, see response to your next comment.
> >
> > W2) "Comparison of different algorithms and design of a new solution approach seems premature"
> >
> > We thank the reviewer for this comment. We agree that the problem formulation is central to our contribution and should be prominent. To address this, we have moved the detailed mathematical formulation of the reward functions (including the asymmetric thaw penalty and carbon accounting) from the Appendix to the main text (Section 3.2). This ensures that the core multi-objective optimization challenge is immediately visible to the reader.
> >
> > Regarding the suggestion to split the work, we agree partially. We agree that the physics of the simulator, the construction of the RL environment, and the baseline algorithmic approaches, together can potentially be split into two works. Which is why we have detailed the physics of the simulator in Appendix B, and are writing a separate paper about it. However, we believe that keeping the environment construction and the algorithmic baselines together strengthens the paper for several reasons. The algorithmic comparison is not intended to claim state-of-the-art algorithmic novelty, but rather to validate the unique properties of the BoreaRL environment. Specifically, the failure of standard preference-conditioned policies and the success of simple site selection strategies demonstrate the asymmetric difficulty between carbon and thaw objectives, a key ecological insight that would be lost without these experimental results. Our analysis shows that the curriculum effectively filters out sites with high potential warming flux (risky sites), allowing the agent to learn on safer, high-latitude sites. See Table 12 in Appendix D.3 for more information on warming fluxes from accepted and rejected sites.
> >
> > A new environment is most useful to the community when accompanied by strong baselines that establish the difficulty of the task. Our results show that BoreaRL is a non-trivial challenge where standard methods fail to maintain preference adherence, thereby motivating future research into robust MORL methods. The "new solution approach" (Curriculum PPO) serves as a naive baseline to demonstrate that domain-aware strategies (like selecting appropriate sites) are necessary for this problem. It sets a performance floor for future researchers to beat, rather than being presented as a standalone algorithmic contribution. We now clarify the wording in the paper to make sure this comes across.
> >
> > We hope that moving the formulation to the main text clarifies the problem definition and retaining the experimental results to demonstrate the environment's complexity and utility as a benchmark makes sense.
> >
> > W3) "Multi-objective metrics"
> >
> > We thank the reviewer for this comment. We agree that scalarized rewards can be sensitive to the relative scaling of objectives. To mitigate this, we implemented domain-informed normalization for both objectives. Carbon reward normalized to [-1, 1] based on a maximum theoretical annual carbon change of 2.0 kg C/m²/yr. Thaw reward: normalized to [-1, 1] based on a maximum annual thaw of 40.0 degree-days.

---

> > > ### Author Response · Authors · 2025-11-27
> > > **Continuing Previous Comment**
> > >
> > > Furthermore, we explicitly address the limitation of scalarized metrics by providing Empirical trade-off coverage analysis (Figure 3d, Appendix D.2). These plots directly visualize the trade-offs and dominance relationships between algorithms, effectively serving the same purpose as multi-objective metrics like hypervolume in demonstrating the superior coverage of MORL approaches.
> > >
> > > However, we acknowledge that *Hypervolume* and *Sparsity* are excellent metrics. We already have Hypervolume as a metric for evaluating the trade-off fronts in our codebase; our analysis suite includes a *compute_hypervolume_2d* function that calculates the hypervolume of the achieved solutions relative to a reference point. We now include explicit discussion about this metric and its results in the paper and use it to quantify the quality of the Pareto fronts generated by different algorithms (see Table 2 in the paper). We now track Sparsity as well, include discussion and results about it, and use it to quantify the uniformity of solutions along the front.
> > >
> > > W4) "Terminology issues"
> > >
> > > We appreciate the reviewer pointing out these terminology issues. We have updated the manuscript to clarify our usage.
> > > We have renamed the captions in Figure 3(d) and Figure 6 to 'Empirical trade-off coverage' and clarified that we plot all weights (averaged across 100 evaluation episodes). This allows us to visualize the range of the learned policies across the preference space, demonstrating the algorithm's robustness, rather than showing only the strictly non-dominated set.
> > > We have added a formal definition in the text for $\lambda$-monotonicity as “the failure of an objective's return to increase with its preference weight”. This metric helps quantify how well the algorithm respects the user's specified preferences.
> > >
> > > Q1) "Reward Design and Validation"
> > >
> > > We have now added more information about our reward components and formulation in the main paper. We also talk about our alternate reward formulation and results. See response to your weakness comment 1.
> > >
> > > Regarding validation, both carbon and thaw rewards are constructed using existing common knowledge from literature about how these fluxes operate. Growth of carbon stock, carbon capacity of forest and soil, overplanting, excessive thinning, etc are common ways to think about the health of a forest in forest management. Thaw degree days and fluxes into and out of the soil are common ways to calculate permafrost thaw. Apart from this existing base, more components can be added to these depending on user preference and is subjective. Therefore, in some sense, these reward formulations are already validated from existing literature. We now mention this explicitly in the paper.
> > >
> > > Q2) "Evaluation Metrics"
> > >
> > > We already had hypervolume in our codebase, but did not include it in the paper. We now discuss and include both hypervolume and sparsity in the paper. See response to your weakness comment 3.
> > >
> > > Q3) "Definitions"
> > >
> > > We now rename this to “Empirical trade-off coverage” and answer this in your weakness comment 4.
> > >
> > > Q4) "Simulator and Environment Use"
> > >
> > > BoreaRL is designed with modularity and extensibility as core principles. The environment adheres to the standard Gymnasium API, making it compatible with almost all modern RL libraries out of the box. The physics engine is encapsulated in the ForestSimulatorclass. It is initialized with a configuration dictionary, allowing users to easily override physical parameters, change time steps, or swap out entire sub-modules (like the weather generator or disturbance models) without rewriting the RL interaction loop.
> > >
> > > As noted in the reward design section, constants like *warming_penalty_factor* and flags like *boreal_thaw_reward_mode* are exposed, allowing users to experiment with different reward shaping functions by simply modifying the configuration. The action space is discrete and users can modify the *_decode_action* method to implement different management interventions (e.g., continuous thinning or different species mixes) with minimal changes to the core logic.
> > >
> > > Q5) "multi-objective learners"
> > >
> > > Yes, we have evaluated BoreaRL with a diverse set of algorithms to demonstrate its utility as a general benchmark. Our codebase includes implementations for scalarized approaches like expected utility policy gradient with fixed weights, preference-conditioned approaches like variable lambda, site selection, which we show improves performance in generalist settings, and gated architectures, which use action masking to handle domain constraints. Our baselines (Fixed-action baselines, Target Density heuristic, Conifer Restoration heuristic) provide non-RL heuristics for comparison.

---

> ### Author Response · Authors · 2025-11-27
> **Continuing Previous Comment**
>
> We plan to add several other multi-objective learning approaches to BoreaRL in the near future (We now mention this as part of future work):
> a) *Evolutionary Multi-Objective Algorithms* like non-dominated Sorting Genetic Algorithm (NSGA-II/III) and Multi-Objective Evolutionary Algorithm based on Decomposition (MOEA/D) that provide population-based alternatives to gradient-based methods.
> b)  *Hypervolume-Based Methods* that can directly optimize trade-off coverage quality metrics.
> c)  *Model-Based MORL* that leverage learned environment models to more efficiently explore the long-horizon, delayed-reward dynamics of permafrost thaw.
>
> Adding these at this stage is beyond the scope of the submitted paper.
>
> In summary, we have re-centered the paper around the problem formulation and reward design you highlighted, and we now treat them as a core contribution rather than background: the detailed mathematical definitions of the carbon and thaw rewards have been moved the main paper, where we explicitly motivate each term, its physical basis, and its ecological meaning, and we clearly explain why thaw is harder to optimize (irreversibility, noisy long-horizon signal, conflicting fluxes) even when the objectives are on comparable scales. To address your request for alternative reward formulations and a more systematic validation, we introduce three thaw objectives (Asymmetric Thaw, Contrast Thaw, Raw Thaw Degree Days), describe their mathematical form and ecological philosophy (risk-averse vs symmetric), and present new results (Table 3 and Appendix D.3) showing that all create genuine conflicts with carbon while the asymmetric form is the most demanding. We also make clear that our design choices are grounded in standard permafrost and ecosystem modelling practice, and we mention this both in the main text and with citations in Appendix B. On evaluation, we have added hypervolume and sparsity metrics on top of scalarized returns and λ-monotonicity, and we explicitly report and discuss these in the main paper (Table 2), directly addressing your concern that multi-objective quality should be measured with standard MORL metrics.
>
> In response to your comments on terminology, we have renamed the “Pareto front” plots to “Empirical trade-off coverage”, clarified that they show all policies across weights (not only non-dominated ones), and we now formally define λ-monotonicity. Regarding the balance between environment and algorithms, we now explicitly state that the algorithmic components are intended as baselines that reveal specific difficulties of the environment (asymmetric objectives, preference adherence, site traps) rather than as a primary novel algorithm; the physics and reward formulation are foregrounded in the main text, while a separate physics-focused paper is in preparation for the simulator itself. Finally, in line with suggestions from you and other reviewers, we have added realistic heuristic baselines, clarified the modularity of BoreaRL (easy switches for reward modes, action spaces, and processes), discussed how the relevance of our work to ICLR and the AI community in general, and expanded the Discussion and Future Work to call out both algorithmic directions and domain-facing directions (calibration, field validation) that BoreaRL now enables.

---

### Official Review · Reviewer_qus2 · 2025-11-06

**Soundness:** 3
**Presentation:** 4
**Contribution:** 2
**Rating:** 4
**Confidence:** 3

**Summary:**

This paper introduces BoreaRL, a multi-objective RL environment for boreal forest management. BoreaRL consists of two components: (1) BoreaRL-Sim, a physics-based simulator running at tunable minute-scale resolution, and (2) BoreaRL-Env, a wrapper around BoreaRL-Sim that implements the mo-gymnasium API and accepts annual agent actions. The “multi-objective” aspect of BoreaRL refers to the dual objectives of carbon sequestration and permafrost preservation. Each episode has 50 time steps, representing 50 years, and the agent’s actions control the tree density as well as ratio between deciduous and coniferous tree species. Experiments across 4 different RL algorithms (fixed-weight EUPG, variable-weight EUPG, PPO Gated, Curriculum PPO) find that Curriculum PPO achieves the best pareto frontier in the multiobjective optimization.

**Strengths:**

**S1) Clarity of writing**

The paper is well-written and generally easy to follow. The motivation is clear, and the conclusions are straightforward to follow.

**S2) Novel multi-objective RL testbed**

The paper presents a novel, and real-world-motivated multi-objective RL problem. Furthermore, the tested algorithms show a large spread in performance, suggesting that the benchmark is useful for testing RL algorithm performance.

**Weaknesses:**

**W1) Questionable claim that permafrost preservation objective is hard to optimize**

A central conclusion from the paper is that the permafrost preservation objective is harder to optimize than the carbon objective. However, I am not fully convinced of this claim. To make this claim stronger, the paper should clarify the scale and range for each of the two reward terms $R_{carbon,t}$ and $R_{thaw,t}$. For instance, if $R_{thaw,t}$ has a range [0, 1] whereas $R_{carbon,t}$ as a range [0,100], then naturally we would expect it to be challenging for an RL algorithm to optimize the permafrost objective.

Also, is there any estimate of what the optimal return might be for the permafrost objective?

**W2) Conclusions can be made stronger with more repeated trials**

It is not clear to me which of the results reported are averaged over multiple runs, vs. a single result. For example, Figure 3b shows error bars, but I am confused by what the error bars are supposed to represent. (Is it showing standard deviation or standard error? Over how many random seeds? And do the random seeds only affect the test episodes, or was the RL algorithm trained multiple times with different random seeds?)

Also, the monotonicity plots (e.g., Figure 3d and elsewhere in the appendix) could benefit from runs with multiple seeds.

**W3) Importance of contribution to AI community is a bit unclear**

While I recognize that BoreaRL is “the first multi-objective reinforcement learning environment for climate-adaptive boreal forest management,” I am not entirely sure how to evaluate the importance of this contribution for an AI conference like ICLR. The paper evaluates several RL algorithms, but it is unclear how BoreaRL compares to other multi-objective RL benchmarks. Suppose some researcher discovers an optimal RL algorithm for BoreaRL; what would be the impact of that?

**Questions:**

Q1) Besides RL, what other tools do forest policy managers traditionally use? Is there any way to compare against such a non-RL baseline?

---

> ### Author Response · Authors · 2025-11-27
> **Clarifying Reward Scaling, Experimental Rigor, and the AI Impact of BoreaRL**
>
> We thank the reviewer for noticing the strengths in our paper and providing constructive comments. We address your comments in order below.
>
> W1. "Questionable that permafrost preservation objective is hard to optimize"
>
> We thank the reviewer for this comment. We agree that clarifying the reward scales is essential to substantiate our claim about the optimization difficulty.
>
> We have updated the *Vector Reward Function* subsection in section 3.2 and *Reward Function* subsection in Appendix C.1 to explicitly state the components of the rewards and normalization details. Both the Carbon and Thaw (Permafrost) objectives are normalized to the range $[-1, 1]$ per step. Carbon is normalized by $2.0$ kg C m$^{-2}$ yr$^{-1}$. Thaw is normalized by $40.0$ degree-days yr$^{-1}$.
>
> Since the numerical scales are comparable, the difficulty in optimizing the permafrost objective arises not from scale mismatch, but from two key factors: a) *Asymmetric Penalty*: The thaw reward function includes a penalty factor $\alpha=2.5$ that penalizes warming (positive heat flux) significantly more than it rewards cooling. This makes the "safe" region of the policy space narrower and harder to find. b) *Conflicting Physics*: The physical mechanisms create a complex optimization landscape. For example, dense canopies reduce summer heat absorption (good for permafrost) but intercept winter snow, reducing insulation and potentially cooling the soil (also good), but the net effect depends on the delicate balance of these opposing forces across seasons.
>
> We have added an estimate that the theoretical optimal return for both objectives over a 50-year episode is approximately $50.0$ (assuming near-perfect performance in every year). The complete details of the reward functions is given both in the main text and in Appendix C.8 “Reward function mathematical formulation”. We have also clarified in the main text that the asymmetric penalty is not arbitrary (In section 3.2 and 4.2). It encodes a *precautionary principle* grounded in the ecological reality that permafrost degradation is often irreversible (tipping point dynamics) and ecologically more damaging than equivalent cooling is beneficial. This makes the objective function inherently risk-averse, which is the right way to frame this objective.
>
> Now, to understand how much of the difficulty comes from the Asymmetric Thaw Objectives, we include and compare 3 alternative thaw objectives: a)  Asymmetric Thaw, b) Contrast Thaw, c) Raw Thaw Degree Days. For more details on these reward formulations and a comparison of their results, see response to Weakness 1 by Reviewer f77E, Table 3 in the paper, and Table 10 in Appendix D.3.
>
> W2) "repeated trials"
>
> We thank the reviewer for this question regarding experimental setup. We have now clarified these details in the revised manuscript.
> Our experimental setup varies between site-specific and generalist modes. In *Site-specific mode*, we train 5 agents with different random site seeds, ensuring robustness to site-specific environmental variations. Each agent is trained independently with a different site configuration. In *Generalist mode*, we train a single agent per method (because we train for longer with each training episode sampled from a random weather and climate seed). The trained agent is then evaluated over 100 episodes per preference weight, where each episode uses a unique random seed. These unique seeds sample diverse weather conditions and site parameters during training and evaluation, ensuring robust performance assessment despite training only one agent.
>
> The error bars in Figure 3b represent the standard deviation (SD) of scalarized rewards computed across the 100 evaluation episodes for each preference weight. This provides a measure of performance variability across different environmental conditions. Figure 3 caption and “Experimental Design” section now explicitly state this. Each point in the monotonicity plots (Fig. 3d and Appendix plots) represents the mean performance over 100 evaluation episodes with unique random seeds. Thus, these plots already benefit from multiple seeds through the training and evaluation procedure described above. The caption now explicitly states this.
>
> We believe these clarifications address the reviewer's concerns about experimental rigor while making our methodology transparent.
>
> W3) "Importance of contribution to AI community"
>
> Thank you for this important question. We believe the contribution of BoreaRL to the AI community is significant for three key reasons:
>
> 1. It presents a novel **Asymmetric Multi-Objective** challenge that breaks standard MORL baselines. You asked: “Suppose some researcher discovers an optimal RL algorithm for BoreaRL; what would be the impact of that?" The immediate impact would be an algorithm capable of solving multi-objective problems where objectives have vastly different signal-to-noise ratios and causal depths.

---

> > ### Author Response · Authors · 2025-11-27
> > **Continuing Previous Comment**
> >
> > In BoreaRL, the Carbon objective is "easy" (direct feedback from planting/growth), while the Thaw (permafrost) objective is "hard" (indirect, noisy feedback driven by complex sub-daily energy fluxes over years). Our experiments show that standard MORL methods (like linear scalarization or simple preference conditioning) fail here. They latch onto the easy Carbon signal and ignore the Thaw signal, or fail to learn at all. An algorithm that solves BoreaRL would likely advance the state-of-the-art in handling asymmetric objectives, a common problem in real-world AI (e.g., optimizing "user engagement" vs. "long-term well-being", or "robot speed" vs. "safety"). BoreaRL isolates this specific difficulty in a reproducible scientific setting.
> >
> > 2. It serves as a testbed for **AI for Science generalization**. BoreaRL is designed from the ground up as a *Generalist* environment. The "Generalist Mode" samples site parameters (latitude, soil conductivity, climate norms) from distributions. An effective policy must learn the *underlying physical relationships* (e.g., "how albedo affects soil temperature") rather than memorizing a specific trajectory. Solving this advances robust and contextual RL, moving us closer to agents that can operate in the messy, uncertain real world.
> >
> > 3. It bridges the gap between toy RL environments and high-fidelity physical simulations. Most “AI for Climate" work relies on either simple toy models or static datasets. BoreaRL provides a **physically-grounded simulator** (solving coupled energy/water/carbon fluxes) that is fast enough for RL training. Discovering optimal policies in BoreaRL generates **scientific hypotheses**. For example, because of our baseline RL agents, we discovered that longer growing seasons (usually good for carbon) can actually help protect permafrost. We stumbled upon this when randomly plotting correlations between variables. We probed the mechanism a bit more and found that longer growing seasons maintain a high Leaf Area Index (LAI) for more days. High LAI increases *Transpiration* (latent heat flux), which consumes energy that would otherwise heat the soil, and increases *Canopy Shading*, blocking solar radiation. We observe a strong correlation between growing season length and thaw protection in our results (see Table 8 in Appendix D.3). This counter-intuitive finding is exactly the kind of hypothesis generation BoreaRL enables. We now discuss this briefly in the paper.
> >
> > Finally, and probably the most important, Boreal forests store ~30-40% of terrestrial carbon. Developing algorithms that can navigate the trade-offs of managing these massive carbon sinks is a high-impact **AI for Good** application, and very relevant to ICLR. Given the existential threat of climate change, afforestation is a critical nature based solution, and environments that enable learning effective, climate-adaptive planting strategies are a crucial contribution to this global effort.
> > In summary, BoreaRL is a specialized instrument for probing specific weaknesses in current RL algorithms (objective asymmetry, physical generalization) while simultaneously addressing a critical climate challenge. We now talk about this in the Introduction and Discussion sections of the paper.
> >
> > Q1. "non-RL baseline"
> >
> > Forest policy managers traditionally rely on a combination of Growth and Yield models (e.g., MGM, GYPSY) to project forest development and Linear Programming (LP) optimization tools (e.g., Remsoft Woodstock) to schedule harvests and silvicultural treatments over long horizons. These tools typically operate on deterministic yield curves and optimize for timber volume or net present value (NPV), often subject to constraints (e.g., even-flow harvest, habitat retention).
> >
> > In our work, we compare against fixed-action baselines (specifically "Zero Density Change" and "+100 Density Increase"), which represent simplified "prescriptive" management regimes analogous to standard silvicultural prescriptions used in forestry (e.g., "plant and let grow" or "continuously densify").  Comparing directly against a traditional LP baseline in this specific context is challenging for two reasons:
> >
> > *Feedback Loops*: BoreaRL models complex, non-linear physical feedback loops between the forest and the local climate (e.g., albedo changes affecting soil temperature, which affects permafrost, which affects growth). Traditional LP models usually assume static yield curves that do not react dynamically to these micro-climatic feedbacks.
> > *Stochasticity*: BoreaRL includes significant stochasticity in weather and climate events, whereas traditional forest planning tools are often deterministic.

---

> > > ### Author Response · Authors · 2025-11-27
> > > **Continuing Previous Comment**
> > >
> > > However, rule-based baselines (e.g., "thin if density > X", "plant if density < Y") could be implemented within BoreaRL to simulate a more complex traditional management strategy. Our current fixed-action baselines effectively serve as the most fundamental version of this, representing a consistent management philosophy applied over time. In response to your comment, we have implemented two additional domain-specific heuristic baselines that mimic standard forestry practices:
> > >
> > > *Target Density Heuristic*: This baseline mimics a standard silvicultural prescription where a manager attempts to maintain a specific stand density (e.g., 1000 stems/ha) by thinning when density is too high and planting when it is too low. This represents a "steady-state" management philosophy often used to maintain forest health.
> > >
> > > *Conifer Restoration Heuristic*: This baseline aggressively plants coniferous species whenever space is available, targeting a 100% conifer fraction. This mimics an industrial forestry approach focused on maximizing timber volume and carbon sequestration, often at the expense of other ecological values.
> > >
> > > We have added these baseline heuristics to our codebase and performed comparisons (see Table 2 in paper). Results indicate that while these heuristics perform reasonably well on single objectives (e.g., Conifer Restoration is strong on Carbon), they fail to navigate the complex trade-offs required for the multi-objective problem. Specifically, the static nature of these heuristics means they cannot adapt to the stochastic site conditions or climate variability in the way that trained RL agents potentially can. For example, both heuristics maintain healthy carbon but fail to protect permafrost because they don't have information about energy fluxes. The fact that our RL agents (particularly Curriculum PPO) outperform these "reasonable human baselines" is evidence that the problem requires the dynamic, state-dependent decision-making capabilities of reinforcement learning and the energy flux information that we compute and compile.
> > >
> > > In summary, we have substantially strengthened and clarified the claims you questioned: we now explicitly define the carbon and thaw reward components, show that both objectives are normalized, so that the relative difficulty of the thaw objective is clearly not an artefact of scaling. To disentangle reward-shaping from inherent physical difficulty, we now introduce and experimentally compare three thaw objectives (Asymmetric Thaw, Contrast Thaw, Raw Thaw Degree Days), which show that agents struggle with thaw across formulations, though most with the asymmetric one; these results, summarized in Table 3 and the new Appendix D.3 tables, directly respond to your concern about robustness of our conclusions with reward design. We have also fully clarified the experimental protocol: the distinction between site-specific and generalist modes, how many agents/seeds are trained, how many evaluation episodes are run per weight, and what the error bars represent.
> > >
> > > Beyond scalarized rewards, we now report standard multi-objective metrics, hypervolume and sparsity, to quantify Pareto-front quality and coverage across methods and weights, and we have renamed and reframed the “Pareto front” plots as “Empirical trade-off coverage”. Finally, in response to your question about non-RL baselines and the broader AI impact, we have implemented two realistic heuristic baselines (Target Density and Conifer Restoration) that mimic standard forestry prescriptions and shown that Curriculum PPO outperforms them on the multi-objective trade-off, and we have expanded the Discussion to better articulate the value of BoreaRL for studying asymmetric multi-objective RL, generalization across physical sites, and climate-relevant AI for Science, incorporating related points raised independently by reviewers f77E, vnbU, and 3zMh.

---

### Author Response · Authors · 2025-11-27
**Cover Note**

We thank the reviewers for their thoughtful and detailed feedback, which has helped us substantially strengthen the paper and the benchmark. Below we summarize the main changes and clarifications made in response to the reviews.

(1) *Reward design, asymmetry, and objective difficulty*. We have added the mathematical formulation of the carbon and thaw objectives from the appendix into Section 3.2, and clarified that both are normalized to $[−1,1]$. We now explicitly justify the asymmetric thaw penalty as encoding the precautionary principle and the irreversibility of permafrost degradation, and we show that the difficulty of the thaw objective is not a trivial scale artifact but arises from noisy, delayed thermodynamic feedbacks. To disentangle reward shaping effects, we added and analysed three alternative thaw rewards (Asymmetric, Contrast, Raw Thaw Degree Days), reporting their behaviour and trade-offs in a new table. This directly addresses concerns about circular reasoning and clarifies what is “domain-imposed” versus “reward-imposed”.

(2) *Experimental rigor and metrics*. Following reviewer suggestions, we incorporated standard multi-objective metrics (hypervolume and sparsity) to quantify Pareto-front quality, and we corrected terminology around “Pareto fronts” by relabelling those figures as “empirical trade-off coverage” and providing a precise definition of λ-monotonicity. We now clearly describe our training/evaluation protocols, number of seeds, and what the error bars in Fig. 3 represent (standard deviation over 100 evaluation episodes per weight). We added tables with per-weight means and variability, and clarified that our trade-off plots aggregate over these 100-episode evaluations.

(3) *Baselines and the need for RL*. To better situate BoreaRL in relation to non-RL practice, we added two domain-informed heuristic baselines (Target Density and Conifer Restoration) that mimic common silvicultural prescriptions. While they perform reasonably on single objectives, they fail on the joint carbon–thaw problem, whereas our RL baselines, especially preference-conditioned PPO with adaptive episode selection, can navigate the trade-off more effectively. This directly addresses concerns about whether the problem truly requires RL.

(4) *Simulator realism and scope*. We expanded Appendix B and the main text to clarify that BoreaRL-Sim assembles standard, validated components from the land-surface and ecosystem-modelling literature (Beer–Lambert radiation, Priestley–Taylor evapotranspiration, LUE+Q10 carbon cycle, multi-layer soil heat conduction as in CLM5/CLASSIC). We corrected the canopy energy-balance equation, added concrete runtime and memory numbers, and discussed how users can extend the simulator (reward hooks, action decoding, configuration interface). We now state more explicitly that BoreaRL is a research benchmark/testbed rather than a ready-to-deploy decision tool, and we outline the calibration and expert-validation steps that would be needed for deployment.

(5) *Positioning and algorithmic claims*. We clarified the contribution of BoreaRL relative to prior environmental RL benchmarks such as CropGym and conservation RL environments: BoreaRL focuses on an asymmetric, physically coupled multi-objective problem with generalization over sites and stochastic weather. We softened and clarified our claims around “Curriculum PPO”: we emphasize that it is a simple preference-conditioned PPO plus adaptive episode selection, offered as a strong, domain-aware baseline rather than an algorithmic centerpiece, and we explain mechanistically why filtering “trap” sites improves preference adherence in this non-convex landscape.

(6) *Scientific hypothesis generation and high-leverage AI-for-good*. We now make explicit that BoreaRL is not only a MORL benchmark but also a tool for generating testable scientific hypotheses about forest management. Learned policies reveal counterintuitive strategies that can be probed in field studies. Because boreal forests store a large fraction of terrestrial carbon and permafrost thaw contributes to potentially irreversible climate feedbacks, advances on BoreaRL target an existential, high-stakes climate problem. We now emphasise that this makes BoreaRL a high-leverage “AI for climate / AI for societal risk” contribution.

Overall, the revised paper (i) makes the core MORL problem and reward structure much more explicit in the main text, (ii) strengthens the empirical section with clearer protocols, additional baselines, and standard multi-objective metrics, (iii) sharpens the positioning of BoreaRL as a physically grounded, asymmetric multi-objective benchmark for AI-for-climate research, and (iv) foregrounds its role in generating scientific hypotheses and enabling high-impact AI-for-good applications. We hope these changes address the reviewers’ concerns and convey more clearly why solving BoreaRL is both algorithmically interesting and societally important.

---

### Meta-Review · Area_Chair_9u3j · 2026-01-09

**Summary:**

This paper introduces multi-objective RL algorithms for boreal forest management, presenting a novel environment leveraging detailed physics-based simulations to evaluate the performance of such methods, and analyzing the effectiveness of different approaches. In particular, the authors consider the challenge of trading off between carbon sequestration and permafrost preservation. The reviewers are positive about much of the work, raising a number of concerns, but overall I believe these have been well addressed by the authors' responses. Accordingly I recommend acceptance.

**Reviewer Concerns:**

These are the substantive reviewer concerns and their status after rebuttals.

Reviewer qus2 (score 4):
- Uncertainty about the purported tradeoff between objectives, in particular the scale of the different objectives. Fully addressed, I believe - e.g., the authors scaled the rewards appropriately.
- Performance under different seeds. Fully addressed - the authors explain their procedure in more detail, showing that they already consider variation across seeds.
- Importance to the ML community. Mostly addressed - I feel the authors indicate why the problem is important + why solving this problem could be aligned with other problems of interest to the RL community.

Reviewer f77E (score 2):
- More attention should be given to how the reward is designed and suggest comparing different rewards. Fully addressed - the authors point to lengthy discussions in the appendices, which they move to the main paper, which show how carefully the reward is designed, based on domain physics. The authors also add experiments comparing different rewards as suggested.
- Could be split into two papers on problem design and algorithm design. Most addressed - I feel this is a judgment call. The authors make an argument against that I personally agree with, and they refocus parts of the discussion per the reviewer's comments.
- Suggest multi-objective metrics for improved evaluation. Mostly addressed - authors introduce a variety of new metrics as suggested.

Reviewer vnbU (score 4):
- Analysis of the performance of algorithms is focused on the actions they pick, not interpreting more deeply what strategies are involved. Mostly address - the authors add discussions on these points.
- Concern that the tradeoff behavior observed is not integral to the application domain, but rather to the way the authors frame it. Fully addressed - the authors explain in detail how the reward design is strongly motivated by the problem space and is therefore integral to it.
- Concerns that the problem isn't novel enough compared to other environmental problems with RL framings. Fully addressed - I agree with the authors that there are very significant differences from other problems considered in the literature, and the argument that "environmental science" is already well-covered in the literature (citing papers in biodiversity and agriculture) is honestly more a sign of the reviewer lumping together very different problem domains rather than a limitation of the present domain.

Reviewer 3zMh (score 4, offers explicitly to increase if concerns even somewhat addressed)
- Is the simulator realistic? Fully addressed, as noted above.
- Human baselines? Mostly addressed - the authors provide a couple of simple ones and discuss them at length including the improvements on them.
- Error bars would be helpful. Fully addressed - the authors added them.

**Reviewer Scores:**

As described above, I feel all the main reviewer concerns are mostly or fully addressed in the rebuttals. I do not see any notable concerns remaining, and recommend the paper for acceptance.

---

### Decision · Program_Chairs · 2026-01-26

Accept (Poster)